# Lysosomal-associated protein transmembrane 5 ameliorates non-alcoholic steatohepatitis by promoting the degradation of CDC42 in mice

Lang Jiang [1,6], Jing Zhao[2,6], Qin Yang[3,6], Mei Li[4,6], Hao Liu[1], Xiaoyue Xiao[1], Song Tian[4], Sha Hu[4], Zhen Liu[5], Peiwen Yang[1], Manhua Chen [2] ✉, Ping Ye [2] ✉ & Jiahong Xia [1] ✉

Non-alcoholic steatohepatitis (NASH) has received great attention due to its high incidence. Here, we show that lysosomal-associated protein transmembrane 5 (LAPTM5) is associated with NASH progression through extensive bioinformatical analysis. The protein level of LAPTM5 bears a negative correlation with NAS score. Moreover, LAPTM5 degradation is mediated through its ubiquitination modification by the E3 ubiquitin ligase NEDD4L. Discovered by experiments conducted on male mice, hepatocyte-specific depletion of *Laptm5* exacerbates mouse NASH symptoms. In contrast, *Laptm5* overexpression in hepatocytes exerts diametrically opposite effects. Mechanistically, LAPTM5 interacts with CDC42 and promotes its degradation through a lysosome-dependent manner under the stimulation of palmitic acid, thus inhibiting activation of the mitogen-activated protein kinase signaling pathway. Finally, adenovirus-mediated hepatic *Laptm5* overexpression ameliorates aforementioned symptoms in NASH models.

Non-alcoholic steatohepatitis (NASH) is pathologically characterized by hepatic steatosis, hepatocyte ballooning, lobular and hepatic inflammation, and interstitial fibrosis[1,2]. As a leading cause culpable for the progression of cirrhosis and hepatocellular carcinoma (HCC), NASH accounts for one in five people with non-alcoholic fatty liver disease (NAFLD) and is estimated to affect approximately 25% of the world adult population according to recent reports[3,4]. Unfortunately, effective therapeutic measures to protect against the development and progression of NASH remain limited, and so far no FDA-approved pharmacological therapy has been available[5,6]. The molecular targets of NASH have attracted mounting attention owing to their good prospect of therapeutic application[7].

In recent years, the role of lysosome-related regulation in disease progression has been a subject of active studies. In fact, lysosomes reportedly not only function to degrade and recycle cellular waste but are key organelles that are implicated in protein degradation, nutrient-sensing, innate and adaptive immunity[8,9]. And so far lysosomes have been proved to interact with multiple signaling pathways and to regulate the progression of a number of diseases, such as atherosclerosis, neurodegeneration diseases, autoimmune disorders, and lysosomal storage disorder. Meanwhile, the protein degradation modulated by the proteostasis system has been believed to be an attractive platform for drug targeting and to play an important role in a wide array of human physiological activities[10,11].

[1]Department of Cardiovascular Surgery, Union Hospital, Tongji Medical College, Huazhong University of Science and Technology, 430022 Wuhan, China. [2]Department of Cardiology, Central Hospital of Wuhan, Tongji Medical College, Huazhong University of Science and Technology, 430014 Wuhan, China. [3]Department of Cardiology, Huanggang Central Hospital, 438021 Huanggang, China. [4]School of Basic Medical Science, Wuhan University, 430071 Wuhan, China. [5]Department of Cardiology, Renmin Hospital of Wuhan University, 430060 Wuhan, China. [6]These authors contributed equally: Lang Jiang, Jing Zhao, Qin Yang, Mei Li. ✉e-mail: cmh_centre@163.com; blue314@163.com; jiahong.xia@hust.edu.cn

The LAPTM family, which consists of LAPTM4A, LAPTM4B, and LAPTM5, have been intensively studied in recent years because of their roles in protein transport and lysosome degradation and can serve as a special target for disease intervention. Lysosomal-associated protein transmembrane 5 (LAPTM5) belongs to the late endosomal/lysosomal transmembrane protein family[12], and was initially identified as a regulator of protein homeostasis[13,14] and a modulator of inflammatory signaling pathways[15]. LAPTM5 improves cardiac hypertrophy by modulating the activity of the MAPK signaling pathway[16]. Our previous research[17,18] and studies by other researchers[19,20] showed that activation of the MAPK signaling pathway was intimately related to the development and progression of NASH. Premised on some preliminary findings, this study was conducted on the hypothesis that LAPTM5 is involved in NASH progression.

In this study, we demonstrated that the protein expression of LAPTM5 was significantly down-regulated in the livers of both human NASH subjects and mouse NASH models. Depletion of LAPTM5 in hepatocytes significantly exacerbated hepatic steatosis, inflammation, and fibrosis in high-fat and high-cholesterol (HFHC) diet-induced mouse NASH models, whereas LAPTM5 overexpression in hepatocytes substantially delayed and mitigated the foregoing pathological changes. We further found that LAPTM5 could directly interact with protein Cell Division Cycle 42 (CDC42) and overexpression of LAPTM5 promoted the lysosomal degradation of it in the circumstance of palmitic acid stimulation. On the other hand, the expression of CDC42 was significantly up-regulated when LAPTM5 expression was decreased, which has been confirmed in both murine and human NASH tissues. As a result, the protective effect of LAPTM5 on lipid deposition and metabolism in hepatocytes and the activation inhibition of the MAPK signaling pathway could be significantly abolished by the over-expression of CDC42. Moreover, hepatocyte lipid deposition due to the knockout of LAPTM5 was significantly suppressed by the CDC42 knockdown, suggesting that LAPTM5 regulates the NASH progression by modulating the protein expression of CDC42 to mediate the activity of the MAPK signaling pathway. Adenovirus-mediated therapy also could considerably ameliorate NASH symptoms. Collectively, these findings revealed that mechanistically LAPTM5 acts as a regulator of NASH progression and clinically it might also serve as an indicator for NASH progression and a target for the treatment of NASH.

## Results

### LAPTM5 expression is down-regulated in fatty liver and correlated with NASH progression

While NASH is pathophysiologically complicated and multifactorial, a large number of proteins have been found to be involved in the regulation of NASH. To know which proteins are the most critical determinants in the pathogenesis of NASH, we searched 10 clinical databases of RNA-Seq from liver samples of NASH subjects and retrieved three conserved proteins, present in lysosomes, certain granules and azurophil granular lumen, that are included in all the 10 clinical databases (Fig. 1a–c). Of note, the severity of the disease is most closely correlated to the expression of proteins localized in the lysosome (Fig. 1d). The search results in 5 databases of RNA-Seq from mouse livers also confirmed this conclusion (Fig. 1e). Given the important role of transmembrane proteins in disease progression[21–23], 71 transmembrane proteins were identified among the above lysosomal associated proteins. High content screening analysis was conducted to evaluate the effect of these genes on lipid profiles and the results showed that LAPTM5 had the strongest inhibitory effect on hepatocyte lipid accumulation upon PA stimulation (Fig. 1f). To investigate the correlation between LAPTM5 and NASH, we firstly determined the protein expression of LAPTM5 in the livers of human subjects without steatosis or with NASH, the hepatic LAPTM5 protein levels were found to be significantly down-regulated in the NASH patients than in their non-NASH counterparts (Fig. 1g and

Supplementary Fig. 1a, b). In combination with the results of immunohistochemistry, we found the protein levels of LAPTM5 were negatively correlated with the NAS score (Fig. 1h, i). In line with our observation in humans, LAPTM5 protein expression was significantly decreased in the livers of *ob/ob* mice and wild-type mice on a high-fat diet (HFD), high-fat high-cholesterol diet (HFHC) or methionine and choline-deficient diet (MCD) (Fig. 1j and Supplementary Fig. 1c–e). Furthermore, in vitro experiments demonstrated that LAPTM5 protein expression was dramatically decreased in a time-dependent manner in both L02 human hepatocytes and mouse primary hepatocytes after PA treatment (Supplementary Fig. 1f, g). Subsequently, the gene expression of *Laptm5* in NASH or non-NASH was detected by the qPCR and the result exhibited, unexpectedly, that the mRNA levels of *Laptm5* were comparable in both in vivo and in vitro models, indicating LAPTM5 was post-transcriptionally regulated in response to metabolic stimulation (Supplementary Fig. 1h-j). Collectively, the striking negative correlation between LAPTM5 expression and NASH development suggests LAPTM5 plays a role in the delayed progression of the condition.

### NEDD4L mediates protein degradation of LAPTM5 through catalyzing its K48-linked ubiquitination

To further explore the mechanism underlying LAPTM5 protein down-regulation in NASH. It was reported that intracellular proteins could be degraded through the ubiquitin–proteasome system or the autophagy pathway[24], inhibitors of the different pathways were treated in hepatocytes stimulated with palmitic acid (PA), and the protein degradation of LAPTM5 was rescued by the proteasome inhibitor MG132, while the lysosome inhibitor Chlq didn't play a role in salvage. (Fig. 2a, b). Then, the proteins that might participate in the degradation of LAPTM5 were IP-mass spectrometrically detected and NEDD4L, NEDD4, WWP2, and ITCH were found to suit the bill (Fig. 2c). The result of mass spectrometry was verified by CO-IP test, and NEDD4L showed the strongest interaction with LAPTM5 (Fig. 2d). Meanwhile, the overexpression of NEDD4L had the strongest promoting effect on the degradation of LAPTM5 (Fig. 2e), suggesting that NEDD4L is a major regulator in LAPTM5 protein degradation. Then, the interaction between LAPTM5 and NEDD4L was further confirmed in vitro by CO-IP and GST precipitation assays (Fig. 2f, g). In order to further understand the mechanism of NEDD4L mediating LAPTM5 degradation, IP assays were conducted to examine the ubiquitination of LAPTM5. While the ubiquitination of LAPTM5 was significantly enhanced when NEDD4L was overexpressed, and this modification was blocked after NEDD4L inactivation (Fig. 2h, i). Our results indicated that the degradation of LAPTM5 was mediated by NEDD4L through K48-linked ubiquitination (Fig. 2j, k). Furthermore, we proved that the degradation was promoted by the E3 ligase of NEDD4L (Fig. 2l). And moreover, the knockdown of NEDD4L successfully rescued LAPTM5 degradation (Fig. 2m). Additionally, we found the protein levels of NEDD4L were significantly up-regulated in the NASH group compared with the control group, which was both verified in the human NASH subjects, mouse NASH models and hepatocytes stimulated with PA (Supplementary Fig. 2a–e). The results proved that NEDD4L was also regulated by NASH.

### LAPTM5 inhibits lipid accumulation and inflammation in hepatocytes

To investigate the effect of LAPTM5 on lipid metabolism and inflammation in hepatocytes, we isolated primary hepatocytes from *Laptm5* knockout (*Laptm5*-KO) mice and *Laptm5*-Flox control mice, with primary hepatocytes infected with adenovirus vector-mediated plasmid over-expressing *Laptm5* (Ad*Laptm5*-Flag) (Fig. 3a and Supplementary Fig. 3a). Nile Red staining (Fig. 3b and Supplementary Fig. 3b) revealed that PAOA-induced hepatocyte lipid accumulation in *Laptm5*-KO group conspicuously deteriorated compared to the control group and

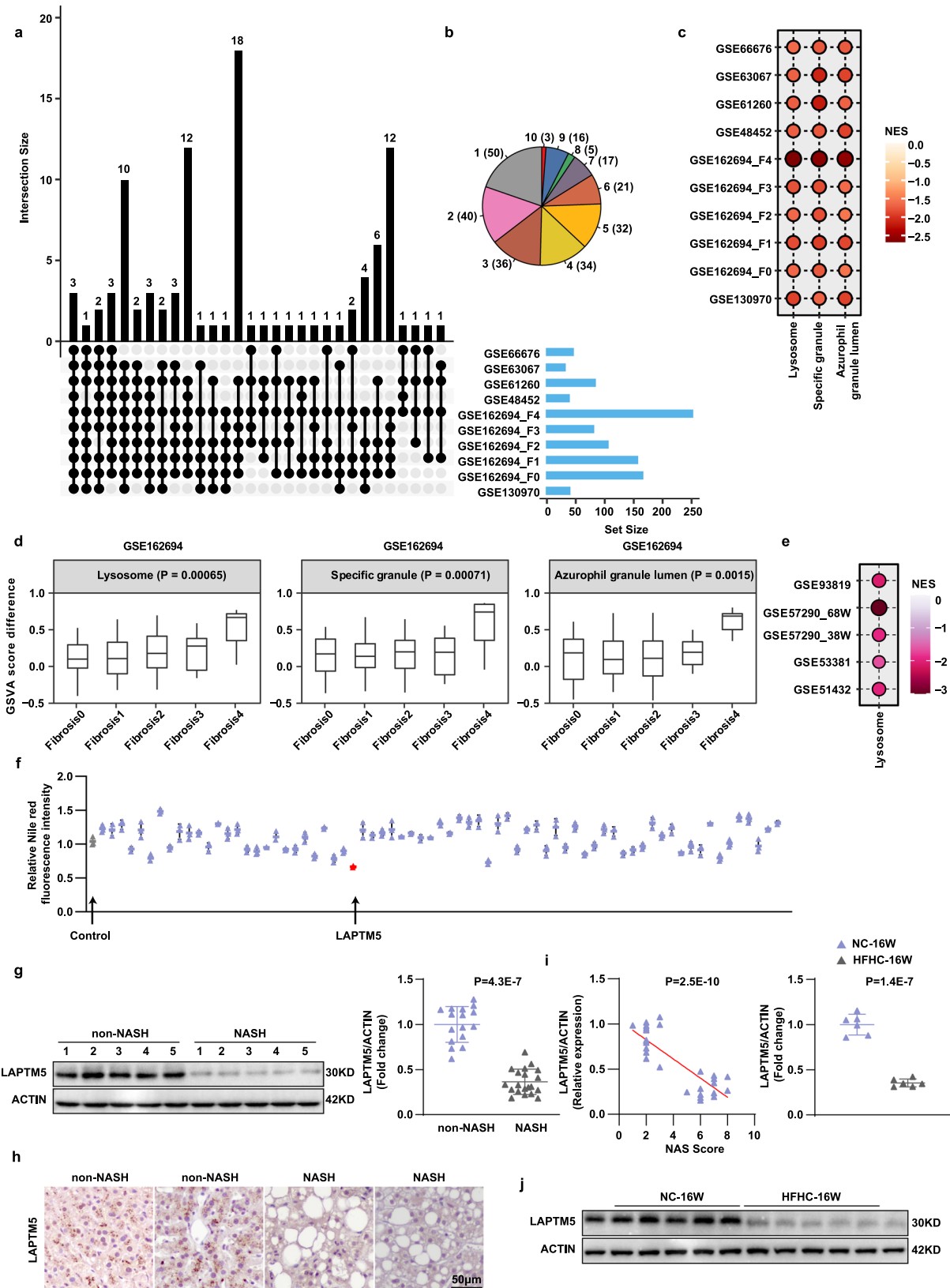

was accompanied by elevated concentrations of triglyceride (TG) and total cholesterol (TC) (Fig. 3b, c). In contrast, *Laptm5* overexpression in hepatocytes ameliorated PAOA-induced lipid deposition (Supplementary Fig. 3b, c). And no significant difference in hepatocytes lipid deposition was observed in BSA-treated groups. Moreover, the inhibitory effects of LAPTM5 on lipid metabolism and inflammation were

further confirmed by qPCR and Western blotting (Fig. 3d–g and Supplementary Fig. 3d–g). Furthermore, on the basis of our RNA-seq data, the hierarchical clustering analysis clearly categorized the PAOA-treated samples into two subgroups: *Laptm5*-Flox-PAOA and *Laptm5*-KO-PAOA (Fig. 3h). It is worth noting that *Laptm5* knockout induced the biological processes that are related to lipid metabolism and

**Fig. 1 | LAPTM5 expression is down-regulated in fatty liver and correlated with NASH progression. a** The GSE derived from RNA-seq of human livers of clinical NASH patients and the health or healthy obesity patients. Genes categories shared between≥10 GSE data are indicated by black dots. The histogram above each plot indicates the times of activated genes categories in each category. **b** The pie chart showed the statistical representation of the genes categories shared GSE. The integers in parentheses represent genes categories and out parentheses represent the counts of GSE shared. **c** NES dots plot of 3 conserved genes category in 10 human databases. **d** GSVA score analysis of these 3 conserved genes in the databases. **e** NES analysis of 5 databases from mouse livers. **f** Quantitative analysis of Nile red fluorescence intensity of L02 cells with 71 molecular overexpression ($n = 3$ independent experiments). **g** Representative western blot analysis (Left) and quantification (right) of LAPTM5 protein levels in the human livers from NASH ($n = 20$ people) or non-NASH group($n = 16$ people). **h** Immunohistochemical staining of LAPTM5 in liver sections of humans in the indicated groups ($n = 5$ people/group). Scale bar, 50 μm. **i** Correlation analysis between LAPTM5 protein levels (normalized to β-actin level) and NAS ($r^2 = 0.6964$, $p < 0.0001$), ($n = 36$ people). **j** Representative LAPTM5 protein levels in the livers from normal chow diet and HFHC diet-fed mice ($n = 6$ mice/group). For (**d**), the data are presented as whisker plots: midline, median; box, 25–75th percentile; whisker, minimum to maximum values; For **f**, **g**, **j**, the data are presented as mean ± SD and two-tailed Student's *t*-test were used for statistical analysis. Source data are provided as a Source data file.

inflammation (Fig. 3i–k). Overall, this in vitro evidence suggests that LAPTM5 exerts a protective effect on metabolic stress-induced lipid accumulation and inflammation in hepatocytes.

### Hepatocyte-specific deletion of *Laptm5* exacerbates steatohepatitis

To further study the influence of LAPTM5 on steatohepatitis and its complications, we constructed hepatocyte-specific *Laptm5* knockout mice (*Laptm5*-HKO) (Supplementary Fig. 4a, b and Fig. 4a) and raised them with normal chow (NC) or HFD diet for 24 weeks. *Laptm5*-HKO mice on a normal diet showed no difference in body weight, liver weight, or lipid profile compared with *Laptm5*-Flox mice. Nonetheless, after 24 weeks of the HFD diet, *Laptm5*-HKO mice exhibited higher liver weight, body weight, fasting blood glucose, and TG/TC levels in the liver and serum than the control group (Fig. 4b–h). Moreover, these measures in *Laptm5*-HKO mice were further exacerbated as compared to the *Laptm5*-Flox mice. Additionally, larger liver and severe lipid accumulation were also observed in the HFD-fed *Laptm5*-HKO mice (Fig. 4i, j), with an expression of genes related to lipid uptake (*Cd36*) and synthesis (*Fasn, Scd1, Pparg*, and *Srebf1*) being up-regulated (Fig. 4k–m). Furthermore, the livers of the *Laptm5*-HKO mice sustained a more serious injury due to the HFD diet, as evidenced by higher alanine aminotransferase (ALT) and aspartate aminotransferase (AST) levels as compared to controls (Fig. 4n, o). These findings revealed that *Laptm5* deletion further drove the NAFLD progression.

Given that NASH is the advanced stage of NAFLD, we fed *Laptm5*-Flox and *Laptm5*-HKO mice with an HFHC diet for 16 weeks to further explore the role of LAPTM5 in a mouse NASH model. Though no difference was found in body weight, the lipid metabolisms indicators, such as liver weight and fasting blood glucose, and lipid deposition aggravated in the HKO group 8 weeks after HFHC feeding, and these indicators were further exacerbated at 16 weeks of HFHC feeding as compared to the Flox group (Fig. 5a–f, and Supplementary Fig. 5a–d). At the same time, the inflammatory infiltration and hepatic fibrosis also deteriorated with protracted HFHC feeding (Fig. 5g–i, and Supplementary Fig. 5e–i). Consistent with the foregoing findings, hepatic *Laptm5* deficiency potentiated the serum levels of ALT and AST (Fig. 5j). Taken together, these results demonstrated that *Laptm5* deficiency significantly aggravated steatohepatitis and its metabolic complications. Then we extracted the mRNA from liver tissues of HFHC-induced *Laptm5*-HKO and Flox mice for sequencing and systematically examined the gene expression profile in the two groups after *Laptm5* deletion in NASH. We found that, in NASH, hepatic *Laptm5* deficiency caused the upregulation of a wide array of pathways and genes that promote lipid metabolism, inflammation, and fibrosis (Fig. 5k–n).

Considering the heterogeneity of NASH, we then evaluated the role of LAPTM5 in a methionine- and choline-deficient diet (MCD)-induced mouse NASH model, and found that inflammatory infiltration and liver damage were substantially more severe[25]. In line with the results of the HFHC-induced NASH model, *Laptm5* deficiency evidently promoted MCD diet-induced liver metabolic disorders and liver injury

(Supplementary Fig. 6a–g). In summary, *Laptm5* depletion aggravates NASH in mice.

### Hepatocyte-specific *Laptm5* overexpression mitigates HFHC-induced NASH

To confirm the role of hepatic *Laptm5* in NASH pathogenesis, we constructed a hepatocyte-specific *Laptm5* transgenic (*Laptm5*-HTG) mice model (Supplementary Fig. 7a and b), with the littermates (NTG) serving as controls. HTG mice exhibited lower liver weight and liver-to-body weight ratios, but no significant change was observed in the body weight as compared to the NTG mice 16 weeks after HFHC feeding. HFHC-induced higher blood glucose and worsened lipid profile were also eased by *Laptm5* overexpression (Supplementary Fig. 7c–g). Furthermore, the HTG mice displayed less severe hepatic steatosis than their NTG counterparts (Supplementary Fig. 7h, i). Coincident with the aforementioned findings, *Laptm5* overexpression greatly mitigated inflammation, fibrosis, and hepatic injury during NASH progression (Supplementary Fig. 7j–p). Collectively, these findings demonstrated that LAPTM5 protects against steatohepatitis and its metabolic complications.

### LAPTM5 suppressed activation of MAPK signaling pathway by promoting the lysosomal degradation of CDC42

To further understand the mechanism underlying the LAPTM5-induced protection against NASH, we integrated the results of RNA sequencing and the Kyoto Encyclopedia of Genes and Genomes (KEGG) pathway analysis and found that *Laptm5* knockout most significantly altered the MAPK signaling pathway (Fig. 6a). Western blotting substantiated that the MAPK signaling pathway was suppressed by *Laptm5* overexpression, but enhanced by *Laptm5* deletion both in vitro and in vivo (Fig. 6b–e). To identify the specific target mediating the suppression of the MAPK signaling pathway, we performed IP-mass spectrometry on *Laptm5*-overexpressing L02 hepatocytes and discovered that LAPTM5 interacted with the small GTP-binding protein CDC42 (Cell Division Cycle 42) (Fig. 6f), which had been reported as a major activator of the saturated fatty acid-stimulated JNK pathway in hepatocytes[26]. We then confirmed that CDC42 overexpression significantly aggravated the lipid accumulation, inflammatory response and promoted the activation of the MAPK signaling pathway in hepatocytes, the finding being consistent with the previously reported results (Supplementary Fig. 8a–e). Subsequently, the interaction between LAPTM5 and CDC42 was further corroborated by CO-IP and GST assay (Fig. 6g, h), and moreover, the interaction was more robust upon PA stimulation (Fig. 6i). Furthermore, the *Laptm5* overexpression inhibited the protein expression of CDC42 under PA stimulation, but had no such effect at BSA condition, the results being verified in both L02 cells and primary hepatocytes (Fig. 6j, k). Whereas *Laptm5* knockout promoted CDC42 expression under PA stimulation, which was also not consistent with the result in BSA condition (Fig. 6l). Then, we further examined the protein levels of CDC42 in the liver tissues from NASH or non-NASH individuals, and CDC42 expression was found to be significantly upregulated in the NASH group, suggesting

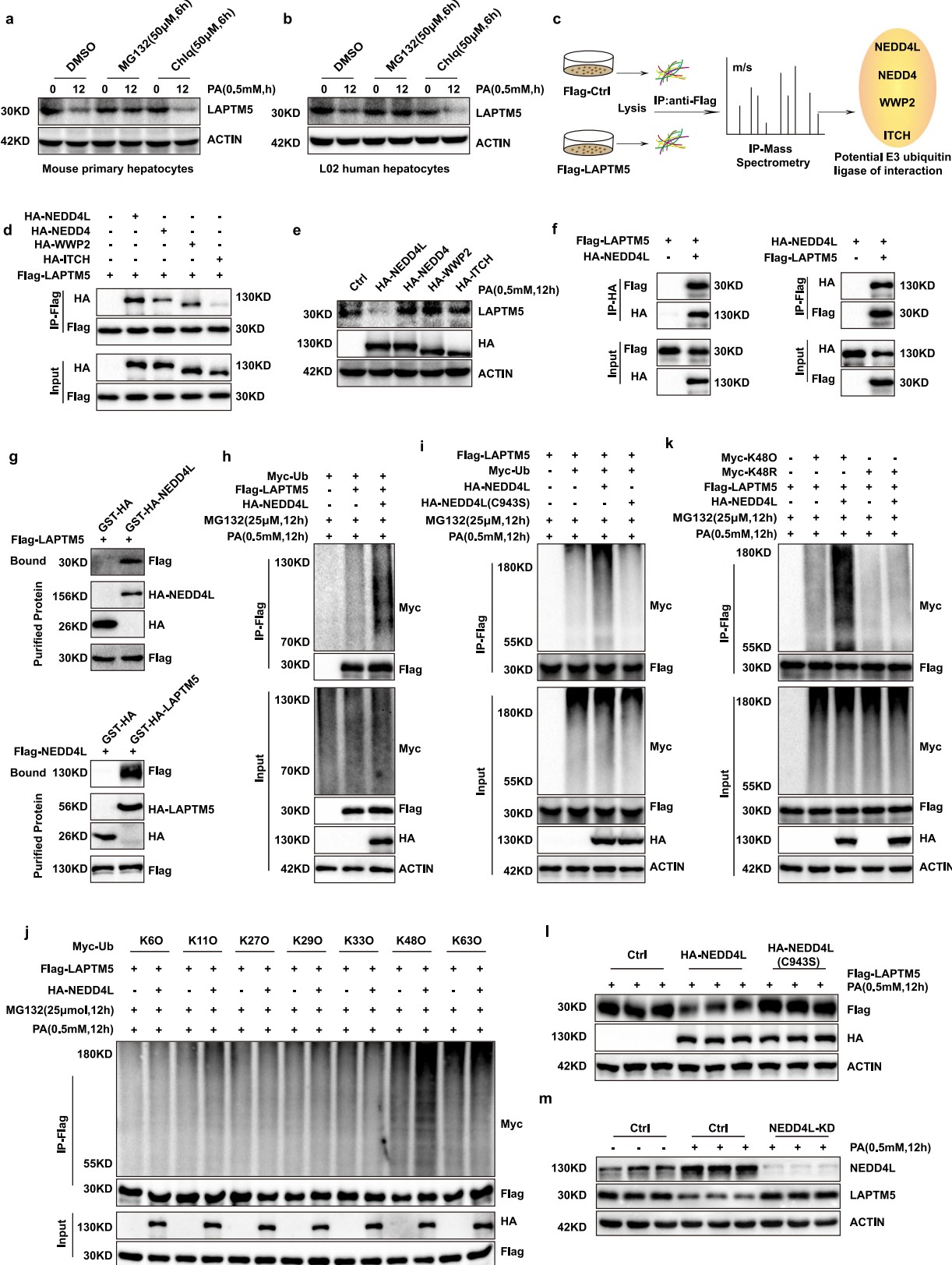

that LAPTM5 and CDC42 were negatively correlated (Fig. 6m), and a NASH-regulating axis existed between LAPTM5 and CDC42. To identify the specific pathway of LAPTM5-mediated CDC42 protein degradation, the cells overexpressing LAPTM5 were treated with both MG132 or Chlq. We found that the lysosomal inhibitor Chlq could abolish the inhibitory effect of LAPTM5 on CDC42 (Fig. 6n, o). Liang et al. and Guo

et al. demonstrated that LAPTM5 could regulate the progression of various systemic diseases, such as tumors and HIV, by promoting the lysosomal degradation of relevant proteins. Therefore, we were led to hypothesize that the down-regulation of CDC42 was mediated by the lysosomal degradation of LAPTM5. What is more, the immuno-fluorescence co-localization staining exhibited that CDC42 was evenly

**Fig. 2 | NEDD4L mediates protein degradation of LAPTM5 through catalyzing its K48-linked ubiquitination. a** and **b** Western blot images of LAPTM5 protein levels in mouse hepatocytes (**a**) and L02 hepatocytes (**b**) treated with MG132 (50 μM), Chlq (50 μM), or DMSO. **c** Procedure of identifying the E3 ubiquitin ligases interacting with LAPTM5 by analyzing IP-MS. **d** Interaction between LAPTM5 and NEDD4L, NEDD4, ITCH, WWP2 in L02 cells. **e** Western blot shows the expression of LAPTM5 after transfected with indicated plasmids. **f** and **g** Co-IP (**f**) and GST pull-down (**g**) show the interaction between LAPTM5 and NEDD4L. **h** Co-IP results show the effect of NEDD4L on ubiquitination of LAPTM5 after MG132 (25 μM) treatment. **i** Co-IP assays of the ubiquitination of LAPTM5 after different treatments. **j** Ubiquitination screening of LAPTM5 by NEDD4L with the indicated types of ubiquitin. (**k**) Ubiquitination of LAPTM5 in L02 hepatocytes transfected with indicated plasmids. **l** LAPTM5 protein levels in L02 cells transfected with indicated plasmids. **m** Western blot results of LAPTM5 expression changes after the knockdown of NEDD4L in primary hepatocytes. Immunoblots are representative of three independent experiments. Source data are provided as a Source data file.

distributed in the cytoplasm under BSA condition (Supplementary Fig. 8f). However, after PA treatment, CDC42 gradually moved toward lysosomes and became granular and eventually co-localized with lysosomes in the cells overexpressing LAPTM5, suggesting that LAPTM5 facilitated the lysosomal endocytic transport of CDC42 and promoted its degradation by lysosomes (Fig. 6p).

### CDC42 mediated the effect of LAPTM5 on lipid deposition and inflammation in hepatocytes

Next, we tried to validate the presence of the LAPTM5-CDC42 regulatory axis in NASH progression. We induced overexpression of CDC42 in *Laptm5*-overexpressing hepatocytes. Our results showed that overexpression of CDC42 could remove the protective effect of LAPTM5 on lipid metabolism stresses, such as the unfavorable lipid profile and up-regulated MAPK signaling pathway when CDC42 was over-expressed (Fig. 7a–d). On the contrary, the knockdown of CDC42 successfully hampered the aggravation of lipid accumulation and inflammation in hepatocytes when *Laptm5* was knocked out (Fig. 7e–h). Collectively, these results indicate that LAPTM5 plays a protective part in NASH by mediating the protein homeostasis of CDC42.

### Adenovirus-mediated hepatic *Laptm5* over-expression alleviated non-alcoholic steatohepatitis

Finally, we examined the therapeutic effect of targeting the LAPTM5-CDC42 axis in NASH. Adenovirus overexpressing *Laptm5* (Ad*LAPTM5*) was injected into mice that had been put on an HFHC diet for 8 weeks. Then, the mice were fed with the HFHC diet for another 4 weeks, AdGFP was used as control (Fig. 8a). Western blot analysis confirmed the LAPTM5 was over-expressed and the expression of p-JNK1/2 and CDC42 was down-regulated in the livers of Ad*LAPTM5*-treated mice (Fig. 8b). Compared to the AdGFP mice, Ad*LAPTM5*-injected mice showed the fasting blood glucose, liver weight, and the liver weight-to-body weight ratio were significantly decreased (Fig. 8c, d). In addition, the Adenovirus-mediated overexpression of *Laptm5* substantially ameliorated lipid accumulation, inflammation, and hepatic injury in the livers of HFHC-fed mice (Fig. 8e–m). Taken together, these findings strongly suggested that LAPTM5 has a good prospect of serving as a therapeutic target for the treatment of NASH and metabolic disorders.

## Discussion

Our results demonstrated that LAPTM5 could be ubiquitination degraded by NEDD4L under metabolic stresses. Overexpression of LAPTM5 attenuated liver steatosis, inflammatory response, and fibrosis. Mechanistically, LAPTM5 can directly bind to CDC42, promotes its lysosomal degradation, and then inhibit the activation of c-Jun NH2-terminal kinase signaling pathway, thus serving its function. Hepatocyte LAPTM5 is a promising therapeutic target for NASH.

LAPTM5 is a multispanning transmembrane protein containing a ubiquitin-interacting motif (UIM) and three PY motifs that bind Nedd4-WW domains. At the same time, the NEDD4-LAPTM5 complex recruits ubiquitinated GGA3 binding to the LAPTM5-UIM[27]. A previous study reported that the E3 ubiquitin ligase ITCH bound to and negatively regulates LAPTM5 through the ubiquitination pathway[28]. Our study showed that LAPTM5 was significantly decreased in NASH models, and

its downregulation was via the ubiquitin-proteasome pathway. Then we found that NEDD4L, and E3 ubiquitin ligases, interacted with LAPTM5 and promoted K48-linked ubiquitination of LAPTM5. Consequently, targeting the physiological regulation loop of NEDD4L-LAPTM5 might be an effective strategy for treating NASH.

The molecular mechanisms underlying the pathogenesis of NASH are multifactorial and intricate[29], multiple studies have shown that the MAPK signaling pathway is involved in the development and progression of NASH. High-fat diet could activate the expression of JNK in the livers[30]. In cellular models of NASH, the saturated fatty acid (palmitic acid) activated PPARα, leading to c-JNK-dependent mitochondrial dysfunction and hepatocyte death[31]. The P38 signaling pathway plays an important role in the inflammation-elicited cell response and stress-induced cell apoptosis[32]. Glowacka et al. and Chen et al. proved that LAPTM5 played a significant role in the modulation and activation of the MAPK signaling pathway[15,33]. In this study, the bioinformatical analysis, in combination with molecular study, showed that the activation of JNK1/2 and p38 was inhibited by LAPTM5 overexpression but enhanced by LAPTM5 deletion in NASH models.

In addition, recent studies documented that the small GTP-binding proteins CDC42 play an important role in the pathogenesis of NASH by modulating the activation of the MAPK signaling pathway[34]. The activation of CDC42 is required for SFA-stimulated MLK3-dependent activation of JNK in hepatocytes, and decreased expression of CDC42 can mitigate the activation of JNK[26]. Consistent with these findings, our study found that LAPTM5 interacted with CDC42 and regulated its expression. Moreover, CDC42 overexpression abolished the protective effect of LAPTM5 on lipid metabolism. LAPTM5 mainly modulated the progression of NASH by regulating the expression of CDC42 and activation of the MAPK signaling pathway.

The interaction between LAPTM5 and CDC42 has yet to be elucidated. LAPTM5 is localized in the lysosomal membrane and involved in a wide array of pathological and physiological processes. Kawai et al. reported that LAPTM5 promoted the lysosomal translocation and degradation of CD3ζ[14]. Ouchida et al. demonstrated that LAPTM5 interacted with BCR and promoted its lysosomal degradation in mouse B cells[35]. In the meantime, several recent studies demonstrated that LAPTM5 could regulate the progression of some systemic conditions, such as malignancies and HIV, by promoting the lysosomal degradation of relevant proteins[36,37]. Additionally, the lysosomal degradation pathway also plays an important role in the progression of NASH. Prior studies have proved that TMBIM1 promoted lysosomal degradation of TLR4 and inhibited high-fat diet-induced insulin resistance, hepatic steatosis, and inflammation[38]. In the present study, we found that LAPTM5 could promote the lysosomal localization and degradation of CDC42 under the stimulation of PA. However, the expression and localization of CDC42 were not influenced under the BSA condition. This indicates that the correlation between LAPTM5 and CDC42 may have changed after the treatment of PA. For example, there may be some changes in the protein domains and molecular activity of LAPTM5 and CDC42 after PA stimulation. Youngshil et al. reported that the protein trafficking and sorting functions of LAPTM5 were strictly regulated by different domains of it[27]. Manju et al. reported that the activation of CDC42 plays a key role in the regulation of disease progression, and the activity intensity of CDC42 also affects the biological

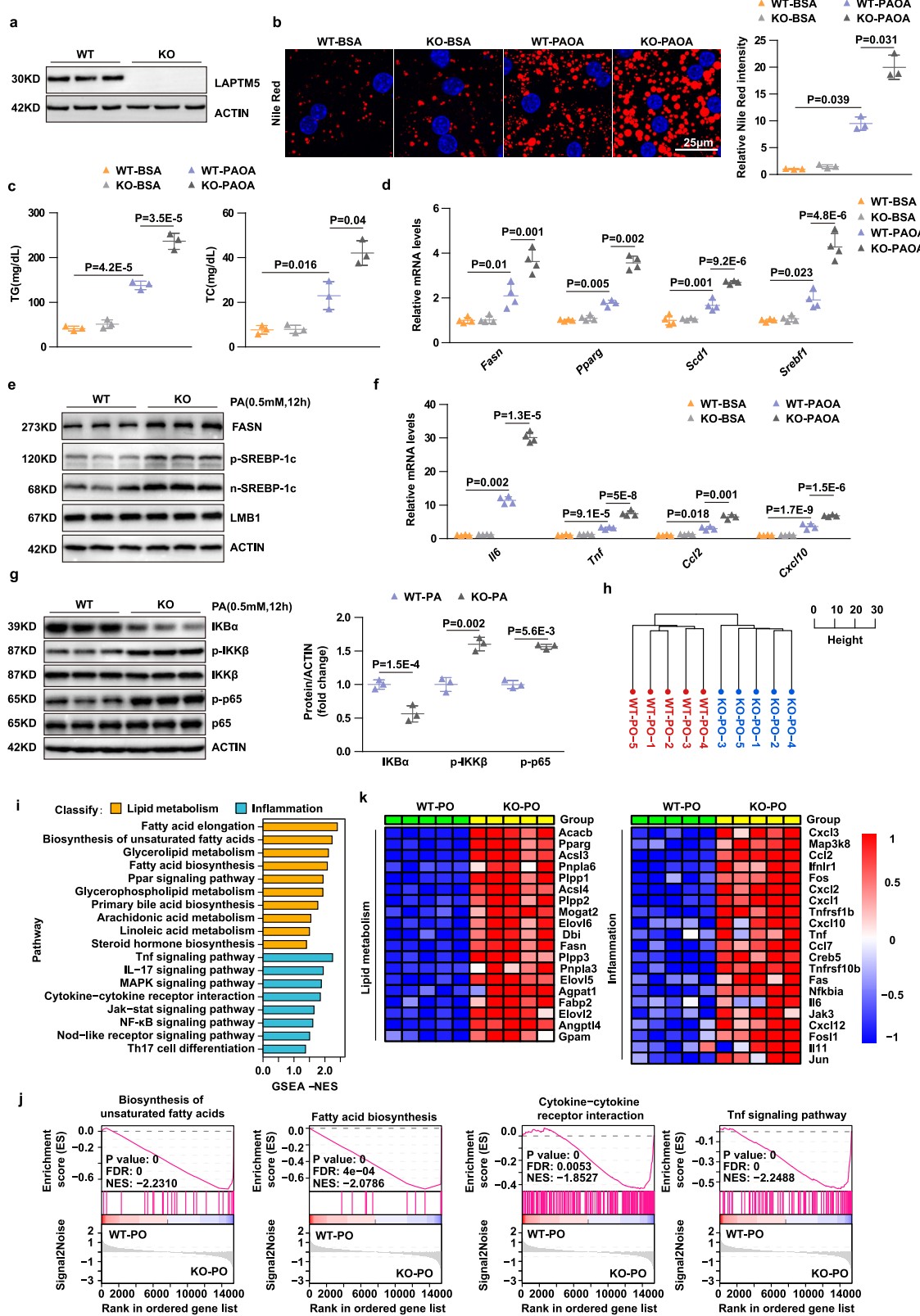

function of it[26]. Therefore, after PA stimulation, delicate and complex changes may have occurred inside the cells. And these deep-seated mechanisms require further study.

In summary, in this study, we identified LAPTM5 as an unreported suppressor of NASH that could negatively regulate the p38/c-JNK pathway by promoting CDC42 lysosome degradation in hepatocytes. Our study showed that adenovirus-mediated *Laptm5* therapy was efficacious against NASH and the LAPTM5 protein expression was correlated with the clinical progression of NASH. These findings indicate that targeting hepatocyte-specific *Laptm5* can be an effective therapeutic alternative for treating NASH and deserves further pre-clinical research.

Some questions concerning the role of LAPTM5 in the treatment of NASH remain to be answered. For example, how does LAPTM5 carry

**Fig. 3 | *Laptm5*-KO exacerbates lipid accumulation and inflammation in hepatocytes. a** LAPTM5 protein levels in hepatocytes isolated from *Laptm5* knockout (KO) mice or WT mice (*n* = 3 mice/group). **b** and **c** Nile Red staining (**b**) and TG, TC contents (**c**) in primary hepatocytes after indicated stimulations. Scale bar, 25 μm, (*n* = 3 independent experiments). **d** and **e** Relative mRNA (*n* = 4 mice/group) (**d**) and protein (*n* = 3 mice/group) (**e**) levels of markers related to fatty acid metabolism in the indicated groups. **f** and **g** Relative mRNA (*n* = 4 mice/group) (**f**) and protein (*n* = 3 mice/group) (**g**) levels of markers related to inflammation in the indicated groups. Protein expression was normalized to β-ACTIN. **h** Hierarchical clustering

analysis of the RNA-seq data from the PAOA-stimulated primary hepatocytes isolated from WT and *Laptm5*-KO mice. (**i-k**) GSEA pathway enrichment analysis and Heatmaps show the activation of pathways and gene expression of lipid metabolism and inflammatory. PAOA, 0.5 mM/1.0 mM; PA, 0.5 mM, OA, 1.0 mM. For all statistical analyses, the results are represented as mean ± SD. One-way ANOVA of Bonferroni post-hoc test (**c**, **d**–*Scd1*, **d**–*Srebf1*, and **f**–*Tnf*) and Tamhane T2 post-hoc test (**b**, **d**–*Fasn*, **b**–*Pparg*, **f**–*Ccl2*, and **f**–*Cxcl10*) was used to evaluate differences. The two-tailed Student's *t*-test was used to evaluate differences in (**g**). Source data are provided as a Source data file.

CDC42 to the lysosome for its protein degradation, and specifically how the process goes? How does CDC42 regulate the activity of downstream p38 and JNK1/2 pathways? All these issues warrant further exploration. Moreover, the correlation between LAPTM5 and NASH remains to be further verified by large-scale clinical trials. Our experiment was conducted in mice and studies involving NASH models of larger animals, such as primates, are needed before moving forward to clinical trials.

## Methods

### Reagents
Key reagents, antibodies, and primers used in this study are listed in Supplemental Materials.

### Mice and treatment
All animal care and related experiments outlined in this study were performed in accordance with the Guidelines for the Care and Use of Laboratory Animals drafted by the US National Institutes of Health (NIH Publication, 8th Edition, 2011). The Institutional Animal Care and Use Committees of Tongji Medical College and the Huazhong University of Science and Technology approved the animal protocols used in this study.

Adult 8–10-week-old C57BL/6 male mice weighing 22–30 g, were housed under pathogen-free conditions in a temperature-controlled environment at 22–24 °C, humidity-controlled environment at 40–70% under a 12-h light/dark cycle with ad libitum access to water and food. Mice were generally in good health before HFD (Protein, 20%; fat, 60%; carbohydrates, 20%; H10060, Beijing HUAFUKANG Bioscience Co., Ltd) for 24 weeks to establish a fatty liver model, HFHC (42% fat, 44% carbohydrate, 14% protein, and 0.2% cholesterol; TP26304; Trophic Diets, Nantong, China) diet for 16 weeks, MCD (TP3005G; Trophic Diets, Nantong, China) diet for 4 weeks to establish the NASH model or NCD (protein, 18%; fat, 4%; carbohydrates, 78%; 1010001, Jiangsuxietong, China), and MCS (TP3005GS; Trophic Diets, Nantong, China). The cervical dislocation was used for mice euthanasia. All animals were fed and bred at the Division of Laboratory Animal Resources of Tongji Medical College.

### Human liver samples
The human sample collection and application were carried out under the supervision of the ethics committee of Renmin Hospital of Wuhan University with the ethics number WDRY2018-K001 and complied with the principles of the Declaration of Helsinki. All individuals have signed an informed consent for the use of clinical specimens in the present study.

We obtained human steatotic liver samples from patients with simple steatosis or NASH who had undergone liver biopsy or liver transplantation. The non-steatotic liver samples were obtained from the healthy regions of the livers of donors who had undergone liver resection because of hepatic cysts or liver hemangioma. Individuals enrolled in this study were excluded from hepatitis virus infection, drug abuse, or excessive alcohol consumption (>140 g for men or >70 g for women, per week). The liver samples from patients are obtained from the non-NASH group of 16 people, and the NASH group

of 20 people. For the non-NASH group, there are 9 female patients and 7 male patients with age ranges from 27 to 67 years old. For the NASH group, there are 13 female patients and 7 male patients with ages ranging from 19 to 40 years old.

### Generation of genetically modified mice
**Construction of liver-specific *Laptm5* gene knockout mice.** *Laptm5*-flox mice were generated using the CRISPR/Cas9 system in C57BL/6 background. Two single guide RNAs (sgRNAs) (listed in Table S1) targeting Laptm5 introns 1 and 2 were designed using an online CRISPR design tool (http://chopchop.cbu.uib.no/). sgRNA expression vectors were constructed using a pUC57-sgRNA backbone (Addgene, 51132). We also designed a donor vector that included two homologous arms, a middle coding region (CDS), and two loxP sequences in the same direction for homologous recombination repair. Next, the sgRNA and Cas9 expression vectors (Addgene, 44758) were transcribed in vitro, and the mixture of mRNAs obtained in vitro with the donor vector was injected into the zygotes of mice using a microinjection apparatus. The injected zygotes were then transplanted into the uterus of recipient mice, and *Laptm5*-flox mice were obtained by genotyping. Subsequently, founder mice were mated with C57BL/6 mice until *Laptm5*-flox/flox mice were obtained and mated with liver-specific Alb-Cre transgenic mice (JAX, 003574). *Laptm5*-flox/flox/Alb Cre mice (*Laptm5*-HepKO), were screened. The identification primers used are listed in Tables P1–P5.

**Construction of liver-specific *Laptm5* transgenic mice.** The CDS region of *Laptm5* was amplified from mouse cDNA obtained by reverse transcription to construct the pALB overexpression vector. Correctly sequenced plasmids were linearized by PvuI restriction endonuclease; the recovered fragments were purified and used for microinjection, subsequently, F0 generation mice were obtained and identified. The first positive mice were mated with the wild type, and the F1 generation positive mice were selected for mating to obtain a stable genetic liver-specific *Laptm5* transgenic mouse strain. The primers used are listed in Table S3.

**Isolation and culture of primary hepatocytes.** Primary hepatocytes were isolated from 6 to 8-week-old male C57BL/6 mice. Briefly, the mice were anesthetized and then the abdominal cavity was opened. The inferior hepatic vena cava was perfused with lavage solution until the liver turned yellow. Finally, the lavage solution containing 0.05% IV collagenase was used to digest the liver. Then the liver was removed, and the liver capsule was opened with microforceps. Next, the tissues were filtered through a 100 μm cell strainer to obtain primary hepatocytes, which were then centrifuged at 50×*g* for 3 min and purified in a 50% Percoll solution. The purified primary hepatocytes were cultured in DMEM containing 10% fetal bovine serum and 1% penicillin–streptomycin in a 5% carbon dioxide/water-saturated incubator at 37 °C.

**Cell lines.** Human hepatocyte L02, human embryonic kidney 293, and 293T cell lines were purchased from the Type Culture Collection of the Chinese Academy of Sciences (Shanghai, China). All the cell lines were

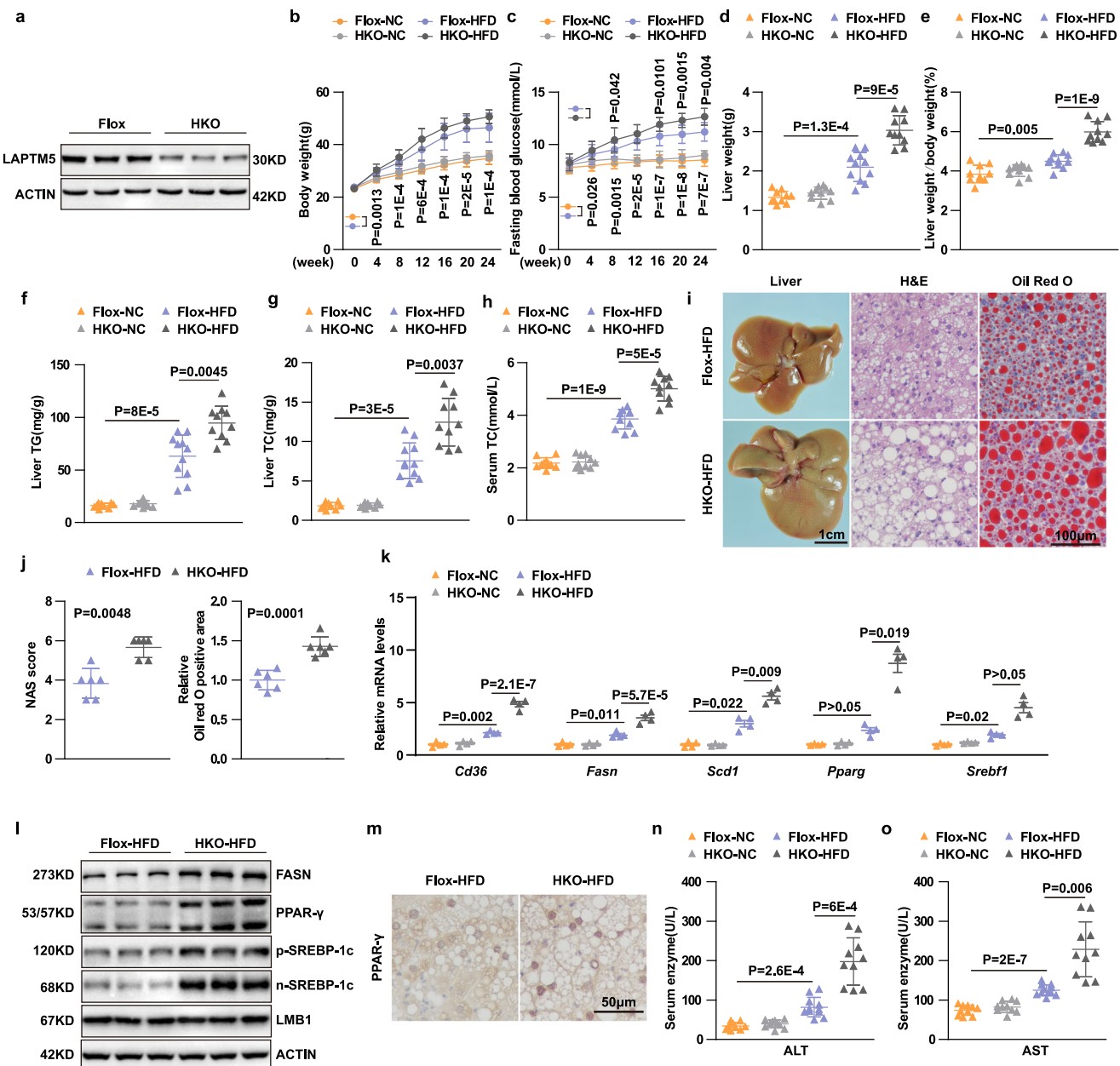

**Fig. 4 | *Laptm5*-HKO exacerbates HFD-induced hepatic steatosis. a** LAPTM5 protein levels in the liver tissues of *Laptm5*-HKO and *Laptm5*-Flox mice (*n* = 3 mice/group). **b** Body weight of *Laptm5*-HKO and *Laptm5*-Flox mice after NC or HFD consumption for 24 weeks. **c–e** Fasting blood glucose (**c**), liver weight (**d**), and ratios of liver weight to body weight (LW/BW) (**e**) of *Laptm5*-HKO and *Laptm5*-Flox mice after NC or HFD consumption for 24 weeks. **f–h** Hepatic TG (**f**), TC (**g**), and serum TC (**h**) content of mice in the indicated groups. **i** Macroscopic and histological images of liver (left, Scale bar, 1 cm), H&E (middle), and Oil Red O (right) (Scale bar, 100 μm) staining of the liver sections of mice in the indicated groups (*n* = 6 mice/group). **j** NAS score analysis (left) and the statistical analysis of Oil red O staining (right),(*n* = 6 mice/group). **k** and **l** Relative mRNA (*n* = 4 mice/group) (**k**) and protein (*n* = 3 mice/group) (**l**) levels of relevant markers in the livers of indicated groups. **m** Immunohistochemical staining of PPARγ in liver sections of mice in the indicated groups (*n* = 6 mice/group). Scale bar, 50 μm. **n** and **o** Serum ALT and AST concentrations in mice in the indicated groups. For **b–h** and **n**, **o**, *n* = 10 mice per NC group, *n* = 11 *Laptm5*-Flox mice and *n* = 10 *Laptm5*-HKO mice for the HFD group. Data are represented as mean ± SD, n.s., not significant. By one-way ANOVA with Tamhane T2 post-hoc test for (**b–d**, **f**–h, k-*Scd1*, k-*Pparg*, and k-*Srebf1*, and **n** and **o**) and Bonferroni's post hoc analysis for (**e**, k-*Cd36* and k-*Fasn*).The Mann–Whitney *U* nonparametric statistical test in (**j**–left) and two-tailed Student's *t*-test in (**j**–right). Source data are provided as a Source data file.

examined for mycoplasma contamination, and the results were negative. The cells were cultured in DMEM containing 10% fetal bovine serum and 1% penicillin–streptomycin in a 5% carbon dioxide/water-saturated incubator at 37 °C. Before the experiments, all cell lines were verified through short tandem repeat DNA profiling. All the cell lines in our laboratory were passaged no more than 30 times after resuscitation and routinely tested for mycoplasma contamination by PCR. To establish a hepatic steatosis model in vitro, mouse primary hepatocytes and

L02 cells were stimulated with palmitic acid (PA; 0.5 mM; P0500; Sigma-Aldrich; St. Louis, MO, USA) and oleic acid (OA; 1.0 mM; O-1008; Sigma-Aldrich; St. Louis, MO, USA) (dissolved in 0.5% fatty acid-free BSA) at the stated concentrations for 12–24 h. For the control group, cells were stimulated with fatty acid-free BSA (0.5%; BAH66-0100; Equitech Bio, Kerrville, TX, USA). To determine whether LAPTM5 facilitates the proteasomal degradation of CDC42, the cells were treated with 50 μmol/L of Chlq (S6999; Selleck Chemicals) for 12 h.

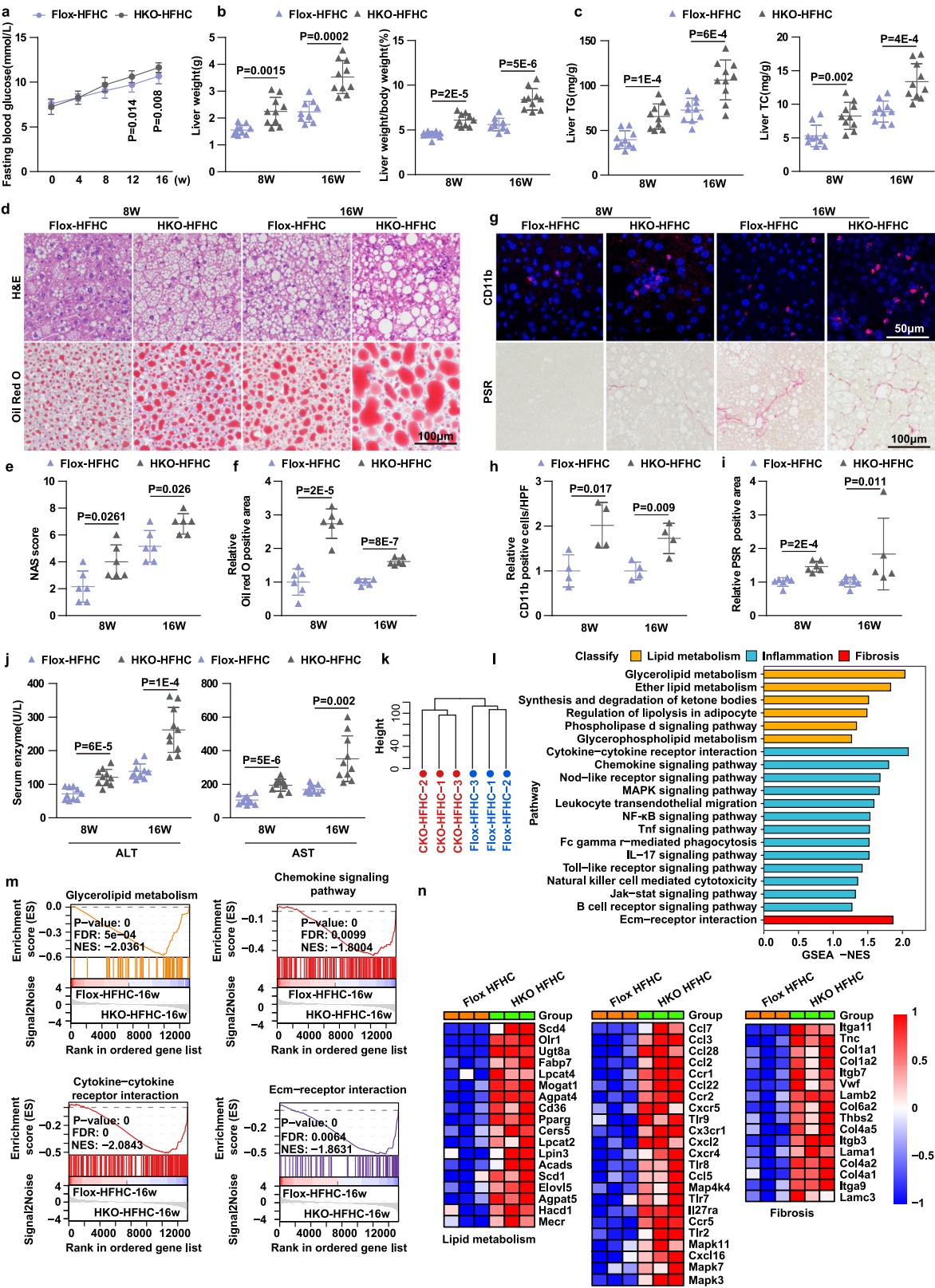

**Mouse metabolic and liver function assays.** The fasting blood glucose levels and body weight levels of the mice were assessed every 4 weeks, and the fasting blood glucose levels were measured using a glucometer (Life Scan, Milpitas, CA, USA). Serum TC, ALT, and AST concentrations were measured using an ADVIA 2400 Chemistry System Analyzer (Siemens, Tarrytown, NY, USA) according to the manufacturer's instructions.

**Lipid analysis.** The TG and TC contents (290-63701 for TG, 294-65801 for TC; Wako; Tokyo, Japan) of primary hepatocytes, L02 cells, and

**Fig. 5 | *Laptm5*-HKO exacerbates HFHC-induced NASH. a** Fasting blood glucose of *Laptm5*-HKO and *Laptm5*-Flox mice for NC or HFHC consumptions (*n* = 10 mice/group). **b** and **c** Liver weight and LW/BW (**b**), and hepatic TG, TC contents (**c**) of *Laptm5*-HKO and *Laptm5*-Flox mice after NC or HFHC feeding for 8 or 16 weeks (*n* = 10 mice/group). **d** H&E (upper) and Oil Red O (lower) staining in the liver sections of mice in the indicated groups (*n* = 6 mice/group). Scale bar, 100 μm. **e** and **f** NAS score analysis (**e**) and the statistical analysis of Oil red O staining (**f**) of *Laptm5*-HKO and *Laptm5*-Flox mice after NC or HFHC feeding for 8 or 16 weeks (*n* = 6 mice/group). **g** Immunofluorescence staining (**g**) and statistical analysis (**h**, **i**) of CD11b (red) in the liver sections of mice in the indicated groups. (Nuclei, blue) (*n* = 4 mice/group). Scale bar, 50 μm. PSR staining of mice liver sections in the indicated groups. (8 weeks, *n* = 6 mice/group, 16 weeks, *n* = 7 *Laptm5*-Flox mice and *n* = 5 *Laptm5*-HKO mice). Scale bars, 100 μm. **j** Serum ALT and AST concentrations of mice in the indicated groups (*n* = 10 mice/group). **k** Hierarchical clustering analysis of the RNA-seq data from the mice fed the HFHC diet. **l** and **m** GSEA pathway enrichment analysis of pathways related to lipid metabolism, inflammation, apoptosis, and fibrosis. **n** Heatmaps of the genes related to lipid metabolism, inflammatory responses, and fibrosis (red, upregulated; blue, downregulated) in the indicated groups. Data are represented as mean ± SD. The Mann−Whitney *U* nonparametric statistical test was used for statistical analysis in (**c**−8w, **e**−16w, and **i**−16w) and two-tailed Student's *t*-test in other panels. Source data are provided as a Source data file.

---

liver tissues were measured using commercial kits according to the manufacturer's instructions.

**Histopathological analysis.** The liver tissues of mice were fixed with 10% formaldehyde for 48 h, and the tissue block was subsequently trimmed and embedded through tissue dehydration. The OCT-embedded liver sections were stained with Oil Red O (O0625; Sigma-Aldrich; St. Louis, MO, USA), and paraffin-embedded liver sections were stained with H&E (Hematoxylin, G1004, Servicebio, Wuhan, China; Eosin, BA-4024, Baso, Zhuhai, China) and picrosirius red (PSR; 26357-02; Hede biotechnology; Beijing, China), which were used for liver fibrosis testing. All the histological images were acquired through a light microscope (ECLIPSE 80i; Nikon, Tokyo, Japan).

**Immunohistochemistry.** Immunohistochemistry for analyzing PPARγ, and CD11b expression was performed on paraffin-embedded liver sections (5 μm) using PPARγ (1:400 dilution, 2435T, Cell Signaling Technology), CD11b (1:4000 dilution, BM3925, Boster; Wuhan, China) and LAPTM5 (1:200 dilution, sc134676, Santa Cruz) antibodies. For antigen retrieval, samples were heated in a pressure cooker for 5 min in pH_6.0 citrate tissue antigen retrieval solution. After cooling, the samples were placed in 3% $H_2O_2$ for 20 min to quench endogenous peroxide activity. The slides were then blocked with 10% BSA for 30 min after washing with PBS. Subsequently, sections were incubated with primary antibodies overnight at 4 °C, after washing with PBS thrice (5 min/wash), incubating sections with Rabbit Two-step Detection Kit (PV-9001, ZSGB-BIO; Beijing, China) according to the manufacturer's protocol. Immunohistochemical staining was visualized using a 3,3′-diaminobenzidine (DAB) substrate kit (ZLI-9018; ZSGB-BIO, Beijing, China) and hematoxylin counterstaining. Images were captured using a light microscope (ECLIPSE 80i; Nikon, Tokyo, Japan).

**Immunofluorescence staining.** To perform CD11b immunofluorescence staining, after antigen retrieval, paraffin sections of liver tissues were first labeled with an anti-CD11b (1:800 dilution, BM3925, Boster; Wuhan, China) primary antibody at 4 °C overnight and then incubated with a fluorophore-conjugated secondary antibody [Alexa Fluor 568 goat anti-rabbit IgG (H + L); A11036; Invitrogen; Carlsbad, CA, USA] at 37 °C for 1 h. The immunofluorescence images were acquired through the fluorescence microscope (BX51; Olympus, Tokyo, Japan).

**Nile Red staining.** L02 cells and mouse primary hepatocytes were treated with PAOA for 0, 12, or 24 h. The cells were subsequently fixed with 4% paraformaldehyde for 15 min and stained with Nile Red (1 mM in PBS; 22190; Fanbo Biochemicals; Beijing, China). Lipid accumulation was visualized and quantified using laser scanning confocal microscopy (TCS SP8; Leica, Wetzlar, Germany) or a High-content Analysis System (Operetta CLS, Waltham, MA, USA).

**Confocal microscopy.** For coverslip staining, the L02 hepatocytes transfected with related plasmids were incubated with mouse anti-LAMP1 and rabbit anti-CDC42 antibodies at 37 °C for 2 h and subsequently labeled with fluorophore-conjugated secondary antibodies. Nuclei in all images were stained with DAPI (S36939; Invitrogen; Carlsbad, CA, USA), and the images were acquired with a confocal laser scanning microscope (TCS SP8; Leica; Wetzler, Germany). Before dye loading, cells were treated with PA (0.5 mM) in the presence of Chlq (50 μM) for 12 h.

**Plasmid construction and viral infection.** The PCR products of lysosomal transmembrane proteins were obtained from cDNA libraries and cloned into corresponding vectors to obtain overexpressed plasmids. Full-length sequences of the human LAPTM5, NEDD4, WWP2, ITCH and CDC42 coding regions were subcloned into pcDNA5-HA, pcDNA5-Flag, pcDNA5-Myc and pcDNA5-GST-HA vectors to generate Flag-LAPTM5, HA-NEDD4, HA-WWP2, HA-ITCH, Myc-CDC42, GST-HA-LAPTM5 and GST-HA-CDC42 recombinant plasmids respectively. Similarly, the full-length sequences of human NEDD4L were inserted into the pcDNA3.1-HA vector to generate HA-NEDD4L, and then HA-NEDD4L(C943S) was amplified. The lentivirus package system comprised two packaging plasmids, pMD2.G and psPAX2, and a target phage-Flag-LAPTM5 plasmid. They were mixed and assembled to infect HEK 293 T cells. After 48 h of infection, the supernatant of HEK 293 T cells was collected to infect L02 cells with polybrene (8 μM; H9268; Sigma, St. Louis, MO, USA) to assist transfection. Puromycin (A1113803; Gibco, Grand Island, NY, USA) was used to generate LAPTM5-overexpression stable cell lines, and lentiviral green fluorescent protein (lenti-GFP) was used as a control.

The shuttle plasmid pENTR-U6-CMV-flag-T2A-EGFP and the ViraPower Adenoviral Expression System (V493-20; Invitrogen; Carlsbad, CA, USA) were used to generate mouse LAPTM5-overexpressing adenoviral vectors. After linearization with PacI (R0547L; NEB, MA, USA), the adenoviral vectors were transfected into 293 cells using polyethyleneimine (24765-1; Polysciences, Warrington, UK) transfection reagent. Cells were harvested after 6−7 days to obtain the initial adenovirus. After three generations of amplification, the LAPTM5 adenovirus was purified by cesium chloride density gradient centrifugation, and the titer was measured using the 50% tissue culture infective dose (TCID50) method. Mouse primary hepatocytes were infected with adenovirus at a multiplicity of infection (MOI) of 50. For in vivo assays, adenovirus expressing mouse LAPTM5 protein or GFP (purchased from HANBIO; Shanghai, China) was injected intraperitoneally 8 weeks after HFHC consumption.

**Western blot analysis.** Total protein was isolated and lysed from animal tissues or cells by RIPA lysis buffer, and proteins were quantified using the Pierce BCA Protein Assay Kit (23225, Thermo Fisher Scientific, Waltham, MA, USA). Equal quantities of the indicated proteins were loaded onto a 10% SDS−PAGE gel and transferred to PVDF membranes. The PVDF membranes were subsequently blocked with 5% skim milk dissolved in TBST and incubated with corresponding primary antibodies overnight at 4 °C, followed by HRP-conjugated secondary antibodies for 1 h. Proteins were then detected using an ECL kit and visualized using the ChemiDoc XRS+ imaging system.

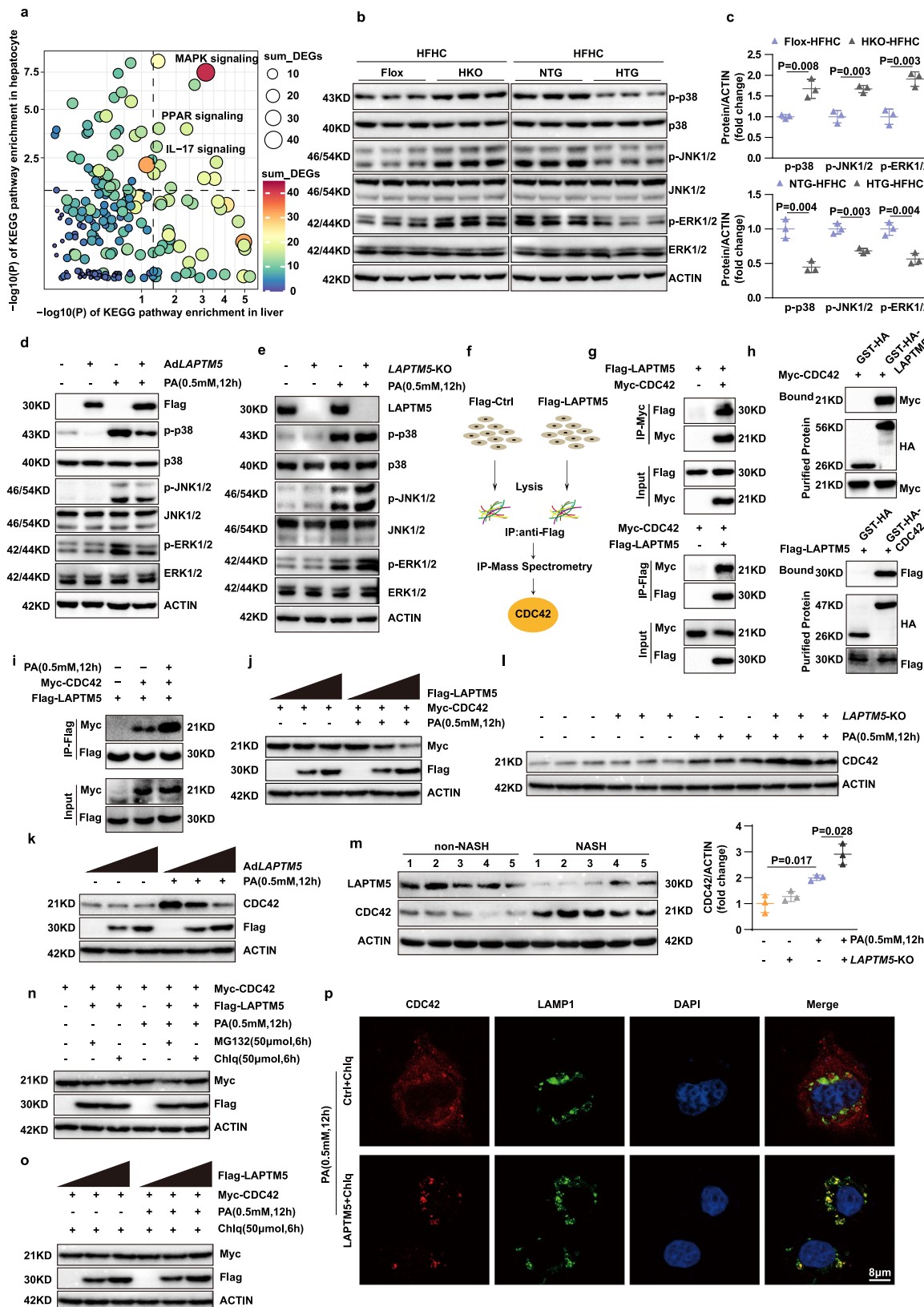

**Quantitative PCR (qPCR) analysis.** Total mRNA was extracted from animal tissues and cultured cells using TRIzol, and cDNA was prepared using the reverse transcription reagents of Vazyme. qPCR was performed using SYBR Green according to the manufacturer's instructions. The mRNA expression levels of the related genes were normalized to that of the housekeeping gene β-actin.

**Immunoprecipitation assays.** Immunoprecipitation (IP) was performed to detect the interactions between proteins. Briefly, HEK293T or L02 cells were cotransfected with the indicated plasmids. After transfection for approximately 16–24 h, cells were lysed using ice-cold 150 mM IP lysis buffer (20 mM Tris–HCl, pH 7.4; 150 mM NaCl; 1 mM EDTA; and 1% NP-40) containing a protease inhibitor cocktail

**Fig. 6 | LAPTM5 suppressed activation of MAPK signaling pathway by promoting the lysosomal degradation of CDC42. a** Combined KEGG analysis results showing the most enriched MAPK pathway. **b** and **c** Western blot images (**b**) and quantitative results (**c**) of phosphorylated and total protein levels of p38, JNK1/2, and ERK1/2 in mice livers of indicated groups ($n = 3$ mice/group). Data represent mean ± SD, two-tailed Student's t-test used to evaluate differences. **d** and **e** Western blot images showing phosphorylated and total protein levels of p38, JNK1/2, and ERK1/2 in the cells from the indicated group. **f** Scheme of identifying the protein interacting with LAPTM5 by analyzing IP-MS. **g** and **h** Co-IP (**g**) and GST pull-down (**h**) shows the interaction between LAPTM5 and CDC42. **i** Co-IP assays for examining the difference in binding strength between LAPTM5 and CDC42 after the stimulation of PA. **j** and **k** Western blot results of exogenous (**j**) and endogenous (**k**) CDC42 expression after overexpression of different concentrations of LAPTM5.

(**l**) Western blot images (up) and Quantitative analysis (down) of CDC42 expression in hepatocytes of LAPTM5-KO mice after PA stimulated ($n = 3$ mice/group). Data represent mean ± SD, and one-way ANOVA of the Bonferroni post-hoc test was used to evaluate differences. **m** Western blot images of LAPTM5 and CDC42 expression in the group of NASH or non-NASH ($n = 5$ people/group). **n** Western blot result of exogenous CDC42 expression after LAPTM5 overexpression under Chlq or MG132 treatment. **o** Western blot images of exogenous CDC42 expression trend with LAPTM5 overexpression in a gradient under the treatment of Chlq. **p** Confocal microscopy images of the co-localization of LAMP1 (green) and CDC42 (red) in L02 cells in the indicated groups (Nuclei, blue). Scale bar, 8 μm ($n = 3$ independent experiments). PAOA, 0.5 mM/1.0 mM; PA, 0.5 mM, OA, 1.0 mM. Chlq, 50 μM; MG132, 50 μM. Immunoblots are representative of three independent experiments. Source data are provided as a Source data file.

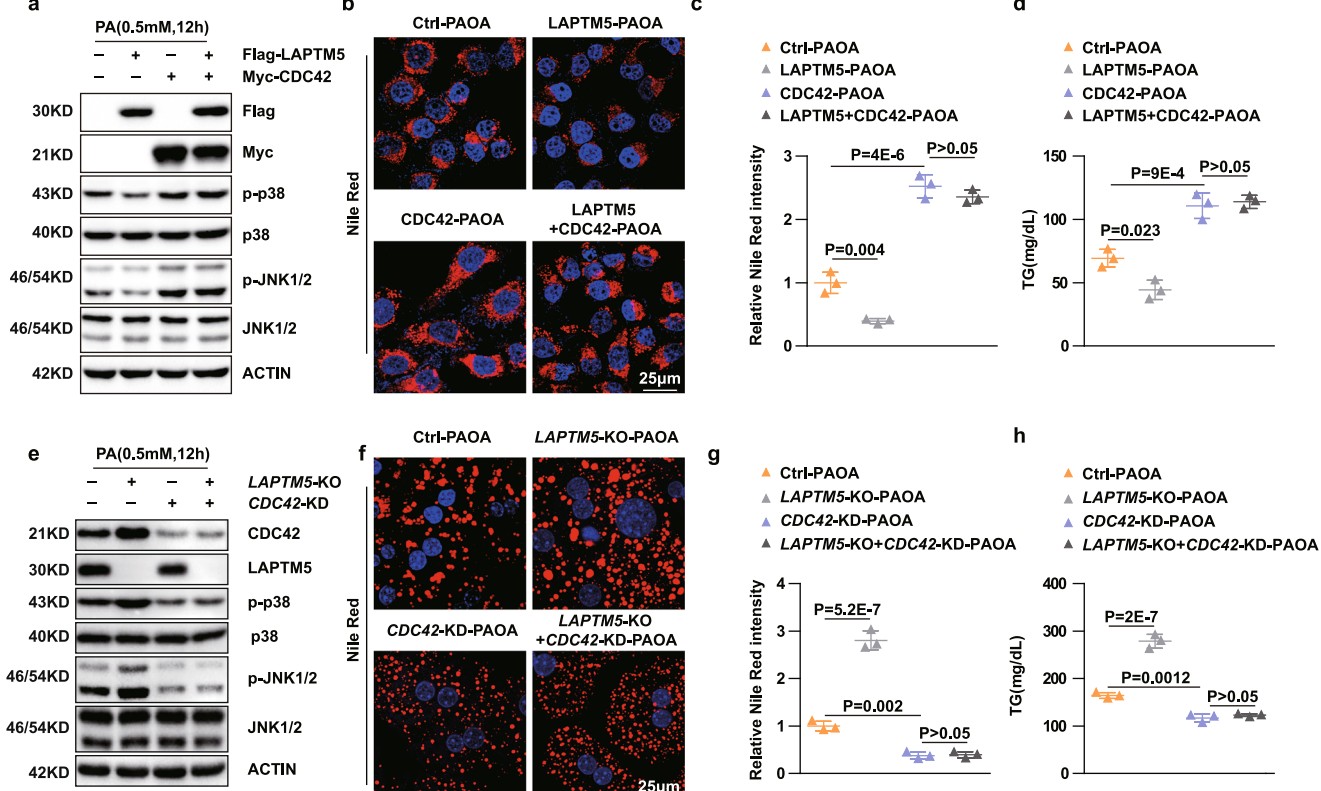

**Fig. 7 | CDC42 mediated the effect of LAPTM5 on lipid deposition and inflammation in hepatocytes. a** Western blot analysis of total and phosphorylation JNK1/2, and p38 after ovexpressing indicated plasmids in L02 cells. **b–d** Nile Red staining (**b**) and TG contents (**d**) of L02 cells after treatment of PAOA in the indicated groups. Scale bar, 25 μm ($n = 3$ independent experiments). **e** Western blot analysis of *Laptm5* knockout primary hepatocytes infected with CDC42 knockdown adenovirus in the indicated groups. **f–h** Nile Red staining (**f**) and TG contents (**h**) of

primary hepatocytes after treatment of PAOA in the indicated groups ($n = 3$ independent experiments). PAOA, 0.5 mM/1.0 mM; PA, o.5 mM, OA, 1.0 mM. Chlq, 50 μM; MG132, 50 μM. Data represent mean ± SD, one-way ANOVA of the Bonferroni post-hoc test was used to evaluate differences in (**c, d**) and (**g, h**). Immunoblots are representative of three independent experiments. Source data are provided as a Source data file.

(04693132001; Roche; Basel, BS, Switzerland). After centrifugation ($12,000 \times g$ for 10 min), the supernatant was collected in a fresh EP tube. Each sample was subsequently incubated with protein A/G agarose beads (AA104307; Bestchrom; Shanghai, China) for 1 h and then with the indicated antibodies at 4 °C overnight. After centrifugation ($3500 \times g$ for 5 min), the beads were washed thrice with 300 mM IP lysis buffer and twice with 150 mM IP lysis buffer. Finally, the beads were heated at 95 °C in SDS loading buffer for 15 min, and separated by SDS–PAGE for western blotting.

**Ubiquitination assays.** L02 cells cotransfected with indicated plasmids were lysed in 80 μL 150 mM IP lysis buffer and 10 μL 10% SDS lysis buffer and then denatured by heating at 95 °C for 15 min. After heating,

900 μL 150 mM IP lysis buffer containing protease inhibitor cocktail was added to the lysates. After sonication and centrifugation ($12,000 \times g$ for 15 min), the supernatant was collected and incubated with indicated antibodies and protein A/G agarose beads for 3 h at 4 °C. The beads were removed after centrifugation ($3500 \times g$ for 5 min) and washed with 500 mM IP lysis buffer (20 mM Tris–HCl, pH 7.4; 500 mM NaCl; 1 mM EDTA; and 1% NP-40) thrice. The beads were then heated at 95 °C with SDS loading buffer for 15 min, and separated by SDS–PAGE for western blotting as previously described.

**RNA-seq and data processing.** For differential gene expression analysis among the different groups, total mRNA was extracted and cDNA libraries were constructed by reverse transcription. Single-end

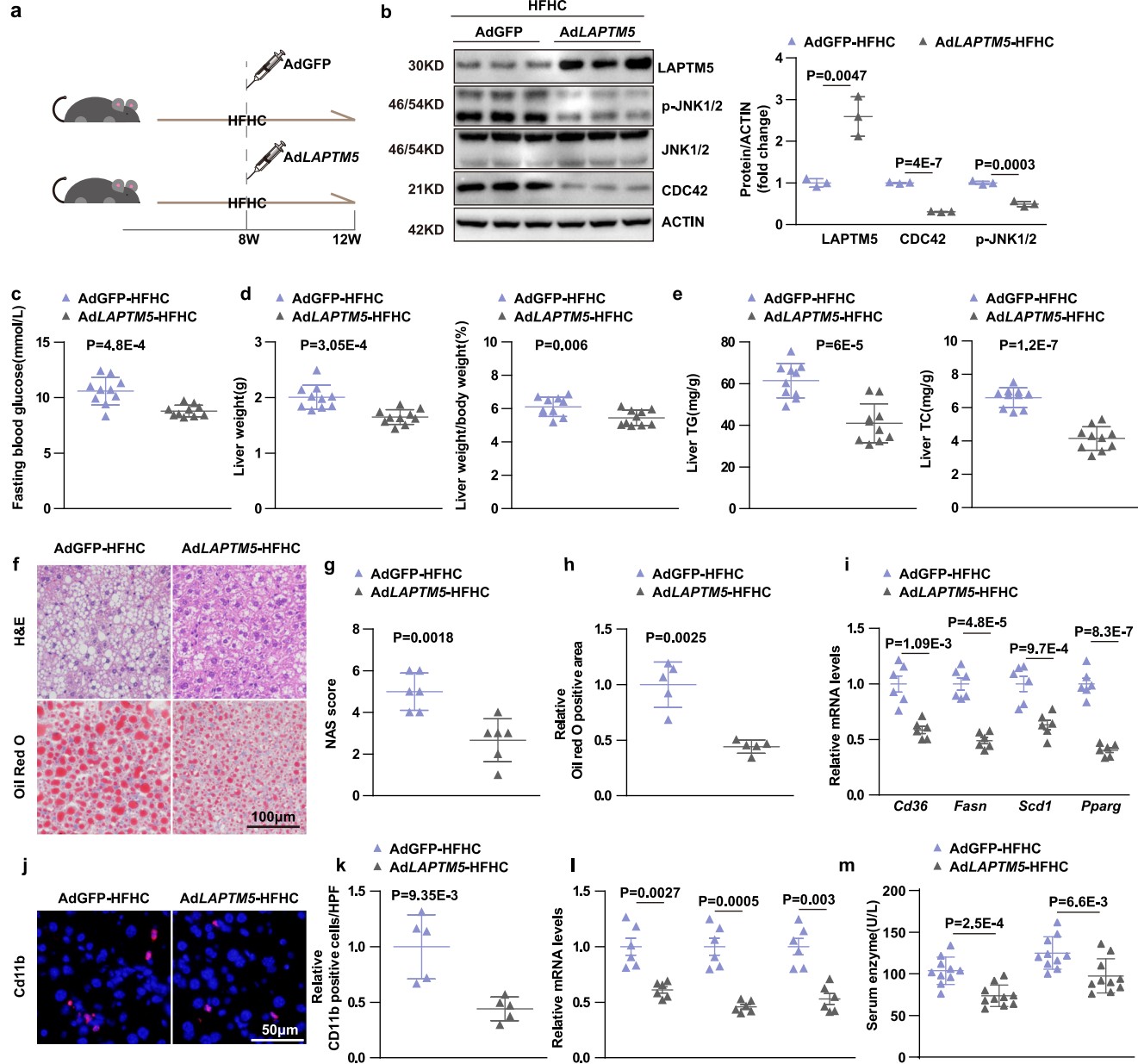

**Fig. 8 | Adenovirus-mediated hepatic *Laptm5* over-expression alleviated non-alcoholic steatohepatitis. a** Scheme of constructing Ad*LAPTM5*-mediated therapeutic NASH models in HFHC mice. **b** Represents the WB detection results of the proteins indicated in the groups (*n* = 3 mice/group). **c** and **d** Fasting blood glucose (**c**), liver weight, and LW/BW (**d**) of mice in the indicated groups (*n* = 10 mice/group). **e** Hepatic TG and TC contents of mice in the indicated group (*n* = 10 mice/group). **f** H&E (upper) (*n* = 6 mice/group) and Oil Red O (lower) (*n* = 5 mice/group) staining in the liver sections. Scale bar, 100 μm. **g** NAS score analysis of the group in panel (**f**) (*n* = 6 mice/group). **h** Statistical analysis of Oil red O in the group of the

panel (**f**) (*n* = 5 mice/group). **i** Relative mRNA levels of genes related to the fatty acid metabolism in the livers of mice in the indicated groups (*n* = 6 mice/group). **j** and **k** Immunofluorescence staining (**j**) and Statistical analysis (**k**) of CD11b (red) in the liver sections of HFHC-fed mice in the indicated groups (*n* = 5 mice/group). Scale bar, 50 μm. **l** Relative mRNA levels of pro-inflammatory genes in the livers of mice in the indicated groups (*n* = 6 mice/group). **m** Serum ALT and AST concentrations of mice in the indicated groups (*n* = 10 mice/group). Data are represented as mean ± SD, two-tailed Student's *t*-test was used to evaluate differences in all panels. Source data are provided as a Source data file.

libraries were sequenced using a BGISEQ 500 (MGI Tech; Shenzhen, China). The reads were mapped to Ensembl mouse (mm10/GRCm38)/human (hg38/GRCh38) reference genomes using HISAT2 software (version 2.1.0). Binary Alignment Map (BAM) files were generated through SAMtools (version 1.4), and fragments per kilobase of exon model per million mapped fragments (FPKM) values of genes were calculated using StringTie (version 1.3.3b); DESeq2 (version 1.2.10) was used for differential gene expression analysis. Genes with a fold change > 1.5 and corresponding adjusted *P* values <0.05 were identified as DEGs.

**Clustering analysis.** We used the unweighted average distance algorithm to generate a clustering tree for the hierarchical clustering analysis. The gene expression levels of each biological replicate were normalized using the *z*-score method.

**Gene set enrichment analysis (GSEA).** Every KEGG pathway or GO biological process term and the involved genes were defined as a gene set, and a ranked list and a '"gene set" permutation type were generated using '"Signal2Noise'" metric to implement GSEA using the Java GSEA (version 3.0) platform.

**KEGG pathway enrichment analysis.** We performed KEGG pathway enrichment analysis using Fisher's exact test with our in-house R script and downloaded KEGG pathway annotations from the KEGG database. At the same time pathways with a $P$ value <0.05 were defined as significantly enriched pathways.

**Mass spectrometry analysis.** We described LAPTM5 and its interacting proteins that were immunoprecipitated in the IP assay. Proteins were subjected to liquid chromatography–tandem mass spectrometry (LC–MS/MS) analysis. The standards for selecting the candidate molecules were as follows: (1) the candidates must be present in the PAOA-treated group but diminished in the BSA-treated group: (2) the number of unique peptides must be >2.

**Statistical analysis.** All data were analyzed through corresponding statistical methods using the SPSS 21.0 software. Differences between two groups with normal distribution were determined using the Student's t-test. For comparisons among three or more groups with normal distribution, one-way ANOVA followed by the Bonferroni post hoc test (for data showing homogeneity of variance) or Tamhane T2 post hoc test (for heteroscedastic data) was applied. When the data met a non-normal distribution, a Kruskal–Wallis nonparametric statistical test was used. Statistical significance was set at $P < 0.05$. All data are presented as mean ± SD values, and the statistical methods used with the corresponding $P$ values for the data are mentioned in each figure legend. We collected data from the animal studies in a blinded manner, and no data were excluded from the final statistical analysis.

### Reporting summary
Further information on research design is available in the Nature Portfolio Reporting Summary linked to this article.

## Data availability
The RNA-seq data generated in this study have been deposited in the National Center for Biotechnology Information Sequence Read Archive (SRA) database under accession code PRJNA918313 (ID 918313 - BioProject - NCBI (nih.gov)) and PRJNA918311 (ID 918311 - BioProject - NCBI (nih.gov)). Processed data of RNA-seq of human livers of clinical NASH patients and the health or healthy obesity patients (accession numbers: GSE66676, GSE63067, GSE61260, GSE48452, GSE162694_F4, GSE162694_F3, GSE162694_F2, GSE162694_F1, GSE162694_F0, GSE130970), RNA-seq of liver samples from mouse NASH (accession numbers: GSE93819, GSE57290_68W, GSE57290_38W, GSE53381, GSE51432) were collected from Gene Expression Omnibus (GEO) database. All the other data supporting the findings of this study are available within the article and its Supplementary Information files and Supplementary tables. A reporting summary for this article is available as a Supplementary Information file. Source data are provided with this paper.

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

## Acknowledgements

We thank Dr. Hao Zhang (Union Hospital, Tongji Medical College, Huazhong University of Science and Technology), Shan-Shan Chen (Central Hospital of Wuhan, Tongji Medical College, Huazhong University of Science and Technology) for their technical assistance and mental encouragement. This work was supported by grants from The Fundamental Research Funds for the Central Universities (HUST No. 2021GCRC037), and the National Natural Science Foundation of China (81730015, 82170504, 81974048).

## Author contributions

P.Ye. and J.X. designed the experiments. L.J., J.Z., Q.Y., and M.L. performed the experiments, data analysis and wrote the manuscript. S.T., S.H., and Z.L. performed bioinformatics analysis. H.L., X.X. and P.Yang. provided technical support. M.C. provided advice and comments. P.Ye., M.C., and J.X. organized and supervised the study.

## Competing interests

The authors declare no competing interests.
