## [Peer Review File · Nature Communications]

Lysosomal protein transmembrane 5 ameliorates non-alcoholic steatohepatitis through degradating CDC42REVIEWER COMMENTS

Reviewer #1 (Remarks to the Author):

Title: Lysosomal-associated protein transmembrane5 ameliorates nonalcoholic steatohepatitis through degrading CDC42

The authors screened LAPTM5 associated with NASH progression through extensive bioinformatics analysis. It was shown that LAPTM5 protein levels negatively correlated with NASH severity. Additionally, the authors demonstrated that LAPTM5 degradation was produced by ubiquitination mediated by the E3 ubiquitin ligase NEDD4L. Depleting LAPTM5 increased hepatic steatosis, inflammation, and fibrosis in experimental NASH. Conversely, LAPTM5 overexpression protects the liver from NASH. The authors concluded that LAPTM5 can be a suitable target for NASH therapeutics and a potential marker of NASH severity. This manuscript is original and results are supported by strong methodology. However, I have some concerns:

- 1.- It is very badly written and therefore difficult to understand. The relevance of the results is not clearly exposed. It seems to me that the manuscript must be rewritten and then edited by high-quality professional service. I will be happy to revise an improved version.
- 2.- One of the main conclusions of the authors is that LAPTM5 inhibits hepatic inflammation. However, the main proinflammatory pathways, namely NF-kappaB and NLRP3 inflammasome signaling pathways were not explored in depth.
- 3.- Hepatic fibrosis was only partially assessed but was not discussed or mechanistically investigated.
- 4.- In general the authors obtained too many results but were not capable of summarizing them nor comprehensively presenting them.

Reviewer #2 (Remarks to the Author):

In the present manuscript, Jiang et al. aimed to address the role of Laptm5 in the development of NASH. The authors found that Laptm5 expression is significantly decreased in both human and mouse livers under steatosis, suggesting a functional relevance. Furthermore, the authors showed that liver-specific deletion of Laptm5 exacerbates fatty liver development and whole body energy homeostasis under either HFD or HFHC feeding, whereas hepatocyte specific overexpression of Laptm5 exerted opposite effect. RNA-seq analysis further showed that pathways related to lipogenesis, inflammation and fibrosis were upregulated by Laptm5 deletion. At the mechanistic level, the authors identified that the MAPK pathway is the most significant and essential downstream pathway of Laptm5 via RNA-Seq and GSEA analysis. Further studies identified that Laptm5 interacted with CDC42 and promoted its lysosomal degradation. Overall, the experiments were well designed and executed. Most of the conclusions can be supported by the data. However, some concerns should be addressed ahead of publication.

Major concerns

1. Laptm family included Laptm4A, Laptm4B and Laptm5, this study revealed the protective function of Laptm5 in the progress of NASH. Whether the other two members, Laptm4A and Laptm4B participated in NASH?
2. The study showed that NEDD4L promoted the degradation of Laptm5, how about the expression of NEDD4L in response to PA stimulation?
3. Whether decreased expression of Laptm5 is mediated by NEDD4L? The rescue experiments are needed.
4. In Figure 3, PO or PA were employed in different experiments, why?
5. In both HFD and HFHC diet induced NAFLD models, the author showed that Laptm5 regulated the expression of genes related to lipid biosynthesis (Fasn, Srebp1, Scd1, Pparg) and transport (Cd36), How about beta-oxidation ?
6. Cd11b and PSR staining were performed to demonstrate the effect of Laptm5 on inflammation and fibrosis. However, it will be more convincing to show more markers of inflammation and fibrosis, such as ly6G, a-SMA etc.
7. In figure 7, the author showed the negative correlation of CDC42 and Laptm5 in mice and primary hepatocytes. More importantly, in my opinion, the author should further confirm this result in clinical samples.

8. In figure 7F, the author identified CDC42 as the interactive protein by IP-MS, was CDC42 the only protein interacted with Laptm5? If not, why the author focused on CDC42? It should be interpreted in detail in your manuscript.

9. This study demonstrated that Laptm5 interacted with CDC42 and promoted its degradation through lysosome manner. However, as a lysosome membrane protein, the mechanism by which Laptm5 promoted lysosome degradation of CDC42 was needed further investigation, or at least be discussed.

10. To investigate whether CDC42 mediated the effect of Laptm5 in response to metabolic stimulation, the author overexpressed CDC42 in Laptm5 overexpression hepatocytes and demonstrated that overexpression of CDC42 blocked the protective effect of Laptm5. However, considering that Laptm5 expression is significantly decreased in clinical samples, it will be more important to further investigate the relationship of Laptm5 and CDC42 in the circumstance of Laptm5 knockdown or deficiency.

Minor concerns

1. Nile red staining (Figure 2B, Figure 2G) should be quantified.

2. There are a lot of typos and grammatical mistakes presented throughout the manuscript. The authors need to carefully verify the language and grammar and the article as a whole should be written in a logical and easy-to-follow manner.

3. There are a lot of redundant data, and the author needs to move them into the supplementary data, such as Figure 3A-B

Reviewer #3 (Remarks to the Author):

In this manuscript, authors identified LAPTM5 as an important regulator of NASH pathogenesis. Authors found that protein level of LAPTM5 was negatively correlated with NAS score, which was mediated by the E3 ubiquitin ligase NEDD4L. Hepatocyte specific depleting LAPTM5 exacerbated hepatic steatosis, inflammation and fibrosis in mouse NASH models. In contrast, LAPTM5 overexpression in hepatocyte exerted opposite effects. Mechanistically, LAPTM5 interacted with CDC42 and promoted its degradation through a lysosomal dependent manner, thus inhibited activation of the mitogen-activated protein kinase signaling pathway. Authors showed lots of interesting data. However, these data were not well connected. There were lots of gaps.

Major

1) It is necessary to determine whether hepatic deletion of Laptm5 increases CDC42 expression. Whether hepatic knockdown of Cdc42 ameliorates the NASH in Laptm5-HKO mice.

2) KEGG analysis of RNA-seq data showed that MAPK signaling pathway was the most significantly altered by LAPTM5 disruption (Fig.7a). Authors should address how LAPTM5 regulates MAPK signaling-related gene expression, and whether this regulation depends on CDC42. Does LAPTM5/CDC42 regulate MAPK signaling at transcriptional level?

3) Hepatic lipid homeostasis is maintained by lipogenesis, FFA uptake, VLDL secretion, FFA β -oxidation. Authors should determine which part is regulated by LAPTM5/CDC42.

4) The molecular weight of each Western blot should be labeled clearly. SREBP-1c immunoblotting was nuclear or cytosolic form? Authors should show both forms of SREBP1c.

5) Many gene expression or Western blot data were collected from HFHC-feeding mice (fig.5e-g, 5i-o) or PAOA-treated hepatocytes (Fig.3k-o). What about NC-feeding mice or non-treated hepatocytes? Is there any difference in lipid metabolism-and inflammation-related genes at these conditions? Data from NC-feeding mice or non-treated hepatocytes will tell readers whether LAPTM5/CDC42 directly regulates gene expression at transcriptional levels.

6) Is NEDD4L also regulated by NASH?

Minor

Authors should cite related papers recently published in NC or other journals.

General response to Reviewers

We sincerely appreciate the reviewers for addressing and reviewing our manuscript. The insightful comments and suggestions greatly helped us to improve the paper and enhanced our understanding on NASH pathogenesis. All concerns raised by reviewers now have been fully addressed by performing extensive experiments, rewriting the manuscript, in-depth analyzing databases and further discussing key points regarding the function and mechanisms of LAPTM5 in NASH progression. The detailed point-by-point responses are listed below, and we hope the revised manuscript and our point-by-point responses are satisfactory.

Point-by-point response to Reviewer #1

Question 1: It is very badly written and therefore difficult to understand. The relevance of the results is not clearly exposed. It seems to me that the manuscript must be rewritten and then edited by high-quality professional service. I will be happy to revise an improved version.

Response: We greatly thank the reviewer for reviewing our manuscript and raising insightful comments and suggestions. Accordingly, we performed additional experiments, rearranged the original figures, and further discussed the key points raised by all reviewers.

As suggested by the reviewer, to make the paper more understandable, we have rewritten the original manuscript and the revised paper has been edited by high-quality professional service. In particular, we clearly described the

rational of this study, emphasized key findings, provided detailed information about methodologies, elaborated all presented results in detail, and discussed the major conclusion and concepts. I hope that the revised manuscript is satisfactory.

Question 2: One of the main conclusions of the authors is that LAPTM5 inhibits hepatic inflammation. However, the main proinflammatory pathways, namely NF-kappaB and NLRP3 inflammasome signaling pathways were not explored in depth.

Response: Regarding the function of LAPTM5 on hepatic inflammation, we have evaluated the hepatic infiltration of inflammatory cells and performed RNA-seq assay to systematically examine the function of LAPTM5 on hepatic inflammatory response. As suggested by the reviewer, we carried out additional experiments and examined the activation of NF-kappaB and NLRP3 inflammasome signaling pathways in the condition of LAPTM5 overexpression or knockout. As shown below and the revised figure 3g-h and supplementary figure 2h-i, the genes expression about NLRP3 inflammasome signaling pathways significantly up-regulated in the condition of LAPTM5 knockout and decreased in the condition of LAPTM5 overexpression. In consistent, the activation of NF-kappaB signaling pathway were inhibited by the overexpression of LAPTM5 but enhanced by LAPTM5 knockout. The results are now added into our revised manuscript (**figure 3g-h & supplementary**

figure 2h-i).

figure 3

supplementary figure 2

Question 3: Hepatic fibrosis was only partially assessed but was not discussed or mechanistically investigated.

Response: Thank you for your suggestion. Hepatic fibrosis is one of the most

important and typical pathological features of NASH, and the degree of liver fibrosis largely reflects the severity of NASH. As the golden standard of NASH diagnosis, histological staining is the most common detection indicators. In our original manuscript, hepatic fibrosis has been examined by PSR staining and further confirmed by RNA-seq assay. Those findings clearly showed that the knockout or overexpression of LPTM5 in hepatocytes could significantly aggravate or ameliorate the progression of NASH, respectively. Furthermore, the RNA-seq-based GSEA assay confirmed that the fibrosis related molecular pathways and gene expression were significantly activated after LPTM5 deficiency (**shown below and revised figure 5I-n**).

In terms of molecular mechanism, about 80-95% of collagen-producing myofibroblasts in different mouse models of fibrosis including NASH were contributed by the activation of hepatic stellate cells (*Gastroenterology*. 2020).

158, 1913-1928.). Thus, to further demonstrate the effect of LPTM5 on hepatic fibrosis and stellate cell activation, we carried out α -SMA staining on liver tissues from *Lap^{tm5}-HKO*, *Lap^{tm5}-HTG* and their corresponding control mice at 16 weeks after HFHC feeding. As shown below, while *Lap^{tm5}* depletion enhanced the infiltration of α -SMA-positive cells in the liver, *Lap^{tm5}* overexpression significantly inhibited HFHC-induced stellate cell activation (**supplementary figure 4c and supplementary figure 5n**). Combined with our results, LPTM5 could be involved in the regulation of fibrosis at both the molecular and phenotypic levels.

Question 4: In general the authors obtained too many results but were not capable of summarizing them nor comprehensively presenting them.

Response: We appreciate the reviewer's suggestions and the recognition of our key findings. To make the manuscript more understandable, as suggested by the reviewer, we have rewritten the original paper and the revised text has been edited by high-quality professional service. We believe that the revised manuscript has been effectively improved, with a clearer description of our key

findings, a detailed description of the methodology and results, and a discussion of key conclusions and concepts.

Point-by-point response to Reviewer #2

Question 1: Laptm family included Laptm4A, Laptm4B and Laptm5, this study revealed the protective function of Laptm5 in the progress of NASH. Whether the other two members, Laptm4A and Laptm4B participated in NASH ?

Response: Thank you for your question. We have conducted relevant experiments to detect the correlation between LAPTMs family and NASH. And we found there's some correlation between LAPTMs4A, LAPTMs4B and NASH, but not as significant as LAPTMs5. The results were listed below.

Question 2: The study showed that NEDD4L promoted the degradation of Laptm5, how about the expression of NEDD4L in response to PA stimulation ?

Response: As suggested, we tested the protein expression of NEDD4L in response to PA stimulation both in mouse primary hepatocytes and L02 human hepatocytes, and the protein expression level of NEDD4L was significantly increased under the stimulation of PA. The results were listed below.

Question 3: Whether decreased expression of Laptm5 is mediated by NEDD4L? The rescue experiments are needed.

Response: Thank you for your valuable question. Our previous experimental results showed that NEDD4L promoted the protein degradation of LAPT5 through the ubiquitination modification at K48 site. And simultaneously, the NEDD4L expression was significantly increased under the stimulation of PA. As your suggestion, we found the degradation of LAPT5 was blocked in the cells of NEDD4L knocking down which proved that the decreased expression of LAPT5 was mediated through the increased expression of NEDD4L with the stimulation of PA. And the results were presented in our revised manuscript (**figure 2p**) as well as listed below.

figure2

Question 4: In Figure 3, PO or PA were employed in different experiments, why ?

Response: Thank you for your question. We apologized for the confusion caused by our lack of clarity. PA (palmitic acid) is one of the important members of the saturated fatty acid family, and it is generally believed that the occurrence and development of NAFLD may be caused by the excessive accumulation of free fatty acids in plasma, especially saturated fatty acids (*Nature metabolism*. 2021. **3**. 1596-1607) (*Hepatology international*. 2020. **14**. 889-919). Therefore, PA is now used to stimulate cell-induced NAFLD disease models in vitro. PO is a mixture of PA and OA (oleic acid) and it was more used in cell phenotyping experiments in figure 3 because the use of PO could make lipid deposition in cells appear more pronounced and make it easier to judge trends in experiments. In other cases, we generally choose to use PA, of course, it is also possible to use PO.

Question 5: In both HFD and HFHC diet induced NAFLD models, the author showed that Laptm5 regulated the expression of genes related to lipid

biosynthesis (Fasn, Srebp1, Scd1, Pparg) and transport (Cd36), How about beta-oxidation ?

Response: We agree with the reviewer that beta-oxidation is a key feature of NASH and should be better examined in the present study. Lipid metabolism is a complex process which contains lipid synthesis, lipid transport and the metabolism process. During our research, we detected the effect of LAPT5 on these processes, and found that lipid synthesis and transport were significantly inhibited by LAPT5. However, the process of beta-oxidation was not significantly influenced as shown below. These data suggested that the beneficial function of LAPT5 against hepatic steatosis might be largely attributed to ameliorate lipid synthesis and transport.

Question 6: Cd11b and PSR staining were performed to demonstrate the effect of Laptm5 on inflammation and fibrosis. However, it will be more convincing to show more markers of inflammation and fibrosis, such as ly6G, a-SMA etc.

Response: Thank you for your valuable suggestion. As reported in the literature, CD11b and Ly6G were classic indicators of hepatic inflammation. In

the meantime, PSR and α -SMA could both represent the severity of liver fibrosis (*J Clin Invest.* 2017. **127(1)**. 55-64.). In our manuscript, CD11b and PSR staining were performed to demonstrate the effect of LAPT5 on inflammation and fibrosis. To further verify the effect of LAPT5 to inflammation and fibrosis, we detected Ly6G and α -SMA staining, and the results were shown in the revised manuscript (**supplementary figure 4c and supplementary figure 5k, 5n**) as well as listed below.

Question 7: In figure 7, the author showed the negative correlation of CDC42 and Laptm5 in mice and primary hepatocytes. More importantly, in my opinion, the author should further confirm this result in clinical samples.

Response: Thank you for your advice. It's very important and meaningful to verify our experiment results in clinical samples. We actively perfected this experiment and the result showed that in the tissues of NASH group, LAPTM5 significantly decreased compared with the non-NASH group, however the expression of CDC42 were significantly up-regulated. And the result was shown in the revised manuscript (**figure 6m**) as well as listed below.

Question 8: In figure 7F, the author identified CDC42 as the interactive protein by IP-MS, was CDC42 the only protein interacted with Laptm5? If not, why the author focused on CDC42? It should be interpreted in detail in your manuscript.

Response: According to the result of IP-MS, CDC42 was not the only protein which interacted with LAPTM5. However, according to the reporting of literature, CDC42 is a major contributor to the saturated fatty acid-stimulated JNK pathway in hepatocytes (*J. Hepatol.* 2012. **56**. 192-198). Therefore, we tentatively studied the correlation of CDC42 with our study and found that CDC42 is indeed interacted with the progression of NASH and it was involved

in the downstream mechanism regulation of LAPTM5. Then we focused on the research of CDC42. Refer to your valuable suggestion, this content of discussion had been added to the results section of figure 6f in our revised manuscript.

Question 9: This study demonstrated that Laptm5 interacted with CDC42 and promoted its degradation through lysosome manner. However, as a lysosome membrane protein, the mechanism by which Laptm5 promoted lysosome degradation of CDC42 was needed further investigation, or at least be discussed.

Response: In our research, through the immunofluorescence staining, we found CDC42 gradually accumulated from the cytoplasm to the lysosomes, and we understand more detailed mechanisms need to be further studied so that to verify our conclusion. Refer to several articles recent published in *Nature Communications*, LAPTM5 could regulate the disease progression in different systemic diseases through promoting the lysosomal degradation of relevant proteins, such as tumor and HIV disease (*Nat. Commun.* 2022. **13**. 4141) (*Nat. Commun.* 2021. **12**. 3691). In recent years, more and more attention had been drawn to lysosomes, which was viewed as an organelle that metabolized cellular waste in the past. However, the function of protein trafficking and degrading modulation of lysosomal transmembrane proteins had been confirmed and extensively documented, which had been recognized as an emerging ways of disease regulation and therapeutic targets (*Nat Rev. Drug Discov.* 2019. **18**.

923-948) (*Immunity*. 2008. **29**. 33-43) (*Nat. Med.* 2017. **23**. 742-752).

Therefore, our result that LAPTM5 interacted with CDC42 and promoted its degradation through lysosome manner could be supported by multiple relevant literatures. However, more in-depth molecular mechanisms inside cells need to be discovered and discussed in our future studies. Refer to your valuable suggestion, this content of discussion had been added to the revised manuscript.

Question 10: To investigate whether CDC42 mediated the effect of Laptm5 in response to metabolic stimulation, the author overexpressed CDC42 in Laptm5 overexpression hepatocytes and demonstrated that overexpression of CDC42 blocked the protective effect of Laptm5. However, considering that Laptm5 expression is significantly decreased in clinical samples, it will be more important to further investigate the relationship of Laptm5 and CDC42 in the circumstance of Laptm5 knockdown or deficiency.

Response: Thank you for your meaningful opinion. Considering that LAPTM5 expression decreased under the stimulation of PA. To further validate that LAPTM5 exerts a protective effect by modulating CDC42, we knockdown the expression of CDC42 in the *Laptm5*-KO hepatocytes. And we found that hepatocyte lipid deposition due to knockout of *Laptm5* was significantly prevented by the knockdown of CDC42. These results confirmed CDC42 mediated protective effects of LAPTM5 on lipid deposition and metabolism in

hepatocytes. We have conducted the experiment and the result was shown in the revised manuscript (**figure 7e-h**) as well as listed below.

Minor concerns:

Question 1: Nile red staining (Figure 2B, Figure 2G) should be quantified.

Response: In the work of article revision, we have carefully quantified every Nile red staining result and performed statistical analysis, and the results were shown in the revised manuscript.

Question 2: There are a lot of typos and grammatical mistakes presented throughout the manuscript. The authors need to carefully verify the language and grammar and the article as a whole should be written in a logical and easy-to-follow manner.

Response: Thank you very much for your advice. We apologize for the mistakes in the manuscript. And we have revised and edited the original manuscript in a logical and easy-to-follow manner by high-quality professional service for better understanding. I hope that the final presentation will satisfy you.

Question 3: There are a lot of redundant data, and the author needs to move them into the supplementary data, such as Figure 3A-B.

Response: Thank you for your valuable advice. We have a lot of experimental results, and we are sorry that we were not able to present the results in the best way. After supplementing the experimental results of requires, we have rearranged all the results to make them more logical and easier to understand in the revised manuscript. And we hope the revised version will satisfy you.

Point-by-point response to Reviewer #3

Question 1: It is necessary to determine whether hepatic deletion of *Laptm5* increases CDC42 expression. Whether hepatic knockdown of *Cdc42* ameliorates the NASH in *Laptm5*-HKO mice.

Response: Thank you for the valuable comments. In our original manuscript, we found that LPTM5 could directly interact with CDC42 and negatively regulate CDC42 protein expression by exogenously expressing LATPM5 and CDC42. Furthermore, the protein expression of LPTM5 and CDC42 was negatively correlated in clinical samples from non-NASH and NASH patients. Here, as suggested by the reviewer, we further examined CDC42 expression in *Laptm5*-KO hepatocytes. In detail, we isolated primary hepatocytes from *Laptm5*-HKO mice and treated them with PA (0.5mM) for 12 hours. After detecting with western blotting, the result showed that hepatic deletion of

Laptm5 significantly increased CDC42 expression. The result was shown in the revised manuscript (**figure 6I**) as well as listed below.

figure 6

Regarding the function of CDC42 in the setting of *Laptm5* deficiency, we isolated primary hepatocytes from *Laptm5*-HKO mice and infected them with CDC42 knockdown adenovirus, followed by subsequent phenotypic detection. The results showed that CDC42 knockdown largely reversed the exacerbated hepatocyte lipid deposition due to knockout of *Laptm5*. In the meantime, the activation of MAPK signaling pathway triggered by *Laptm5* deletion were consistently abolished by the knockdown of CDC42. These data clearly showed that the pro-NASH capacity of *Laptm5* depletion is largely dependent on CDC42 activation. Due to limited time for the manuscript revision, it's hardly to construct the CDC42-KO mice in the background of *Laptm5*-HKO mice, and thus the requirement of CDC42 in LAPTM5-regulated NASH progression was unable to be validated *in vivo*. The *in vitro* results are now added in the revised manuscript (**figures 7e-h**) as well as listed below.

Question 2: KEGG analysis of RNA-seq data showed that MAPK signaling pathway was the most significantly altered by LPTM5 disruption (Fig.7a). Authors should address how LPTM5 regulates MAPK signaling-related gene expression, and whether this regulation depends on CDC42. Does LPTM5/CDC42 regulate MAPK signaling at transcriptional level ?

Response: Thank you for your question. According to the results of **figure 6b-e**, western blotting substantiated that the MAPK signaling pathway was suppressed by *Lptm5* overexpression, but enhanced by *Lptm5* deletion both *in vitro* and *in vivo*. Subsequently, as the results showed below (**figure 7a-h**), overexpression of CDC42 could remove the protective effect of LPTM5 on lipid metabolism stresses, such as the unfavorable lipid profile and up-regulated MAPK signaling pathway when CDC42 was over-expressed (**Fig. 7a-d**). On the contrary, knockdown of CDC42 successfully hampered the aggravation of lipid accumulation and inflammation in hepatocytes when *Lptm5* was knocked out. Which proves that the regulation of LPTM5 to MAPK signaling pathway was depended on CDC42. And according to the report of literature (*Cell Rep.* 2022. **39(1)**. 110641)(*Genes.* 2022. **13(3)**. 468) CDC42 had been proved to play a

significant role in regulating MAPK signaling pathway.

However, whether LAPT5/CDC42 regulate MAPK signaling at transcriptional level, the result of RNA-seq (**figure 5l-n**) has verified that MAPK signaling pathway was significantly regulated after *Lap5m5* being knockout. And we wish our answer and results can satisfy you.

Question 3: Hepatic lipid homeostasis is maintained by lipogenesis, FFA uptake, VLDL secretion, FFA β -oxidation. Authors should determine which part is regulated by LAPT5/CDC42.

Response: We thank the reviewer's comments and we performed additional qPCR experiments to further explore the function of LAPT5 on detailed lipid homeostasis events, including lipogenesis (*Fasn*, *Scd1*, *Pparg*, and *Srebf1*), FFA uptake (*Cd36*) and FFA β -oxidation (*Ppara*, *Cpt1a* and *Acox1*). As shown below, the indicators of FFA β -oxidation exhibited no statistical significance between the groups, whereas LAPT5 significantly inhibited the expression levels of

genes related to lipogenesis and FFA uptake. Consistently, the expression of lipogenesis and fatty acid uptake genes were markedly enhanced by LAPTM5 knockout in the liver of mice after 16 weeks of HFHC treatment. The results were shown in the revised manuscript (**figure 4k, supplementary figure 4d-e, supplementary figure 5i**) as well as listed below.

supplementary figure 4

supplementary figure 5

Question 4: The molecular weight of each Western blot should be labeled clearly. SREBP-1c immunoblotting was nuclear or cytosolic form? Authors should show both forms of SREBP1c.

Response: As suggested, the molecular weight of each Western blot had been labeled in the revised manuscript which had been uploaded as the supplementary files. And the SREBP-1c immunoblotting showed in our

manuscript was cytosolic form, we are sorry that we didn't present both forms before. We later supplemented the experiments and the results are listed below.

Question 5: Many gene expression or Western blot data were collected from HFHC-feeding mice (fig.5e-g, 5i-o) or PAOA-treated hepatocytes (Fig.3k-o). What about NC-feeding mice or non-treated hepatocytes? Is there any difference in lipid metabolism-and inflammation-related genes at these conditions? Data from NC-feeding mice or non-treated hepatocytes will tell readers whether LAPT5/CDC42 directly regulates gene expression at transcriptional levels.

Response: Thank you for your suggestions. We accordingly performed additional experiments to evaluate the influence of LAPT5 on lipid metabolism and inflammation under physiological conditions. In particular, qPCR assay have been carried out using livers collected from *Laptm5*-Flox and *Laptm5*-KO mice after 24 weeks of NC feeding as well as the wild type and *Laptm5*-KO hepatocytes treated by BSA for 12 hours. Notably, we did not observe any significant influences of LAPT5 on those detected genes, suggesting that the function of LAPT5 on NASH are stress dependent. The results were shown in

the revised manuscript (**figure 3d, 3f, 4k**) as well as listed below.

Question 6: Is NEDD4L also regulated by NASH?

Response: Thank you for your question. We tested the protein expression of NEDD4L in response to PA stimulation both in mouse primary hepatocytes and L02 human hepatocytes. As shown below, the protein expression level of NEDD4L was significantly increased under the stimulation of PA. The upregulated NEDD4L is consistent with the reduced LAPTM5 protein expression and the promoting effect of NEDD4L on LAPTM5 ubiquitination degradation

observed in our original study. The results were listed below.

Minor concerns:

Question 1: Authors should cite related papers recently published in NC or other journals.

Response: As suggested by the reviewer, we have updated references in the revised manuscript and cited papers recently published in *Nature Communications* and other Nature series journals, e.g. (Nat. Commun. 2022. **13**. 4141) (Nat. Commun. 2021. **12**. 3691).

REVIEWER COMMENTS

Reviewer #1 (Remarks to the Author):

Thank you for improving the manuscript by performing additional experiments and rewriting it. To me, now it is suitable for publication.

Reviewer #2 (Remarks to the Author):

In the revised manuscript, all the concerns I previously raised have been well addressed. Now the manuscript can be accepted for publication.

Reviewer #3 (Remarks to the Author):

Although authors did some experiments to address my concerns, I was not satisfied with some of the authors' response.

1) Authors did experiments showed that hepatic deletion of Laptm5 increases CDC42 protein levels at PA treated conditions (Fig. 6l). What about normal condition? Authors should show results at both normal and PA treated conditions not only for Fig. 6l but also for Fig. 6j-k, 6n-p. Readers and this reviewer want to see whether LAPTM5 can decrease CDC42 without PA treatment. My second concern "Whether hepatic knockdown of Cdc42 ameliorates the NASH in Laptm5-HKO mice" is important. Authors should address this concern in vivo.

2) Authors did not address my concern "how LAPTM5 regulates MAPK signaling-related gene expression, and whether this regulation depends on CDC42". Authors should list MAPK signaling-related genes regulated by LAPTM5 and CDC42, and performed RT-qPCR assay.

3) Authors did not address my concern "The molecular weight of each Western blot should be labeled clearly. SREBP-1c immunoblotting was nuclear or cytosolic form? Authors should show both forms of SREBP1c". There are still no clear labeling in Figures 3e, 4l, s2e.

4) Authors did not address my concern "Is NEDD4L also regulated by NASH?". Authors should measure NEDD4L protein levels in NASH livers from human patients and mouse model.

General response to Reviewers

We sincerely appreciate the reviewers again for their work in reviewing our revised manuscript. And we are sorry for not being able to fully understand the comments and suggestions so that some of the concerns raised by reviewers were not fully addressed. And now we have performed extensive experiments, labeled the molecular weight of each western blot and rearranged the figures to make our research more rigorous and easier to understand. The detailed point-by-point responses are given below, and we hope the revised manuscript and our point-by-point responses can meet your requirements.

Point-by-point response to Reviewer #1

Remarks to the author: Thank you for improving the manuscript by performing additional experiments and rewriting it. To me, now it is suitable for publication.

Response: Thank you so much for your support of our research, and the valuable comments during the review process. Your suggestions helped a lot in our effort to improve the research and also brought the shortcomings to our attention that helped us revise the manuscript. We have reorganized the results to make them more logical and rewritten the manuscript for better understanding. We are genuinely honored to have you as a reviewer for our study, and we sincerely hope that our study will be published.

Point-by-point response to Reviewer #2

Remarks to the author: In the revised manuscript, all the concerns I previously raised have been well addressed. Now the manuscript can be accepted for publication.

Response: Thank you very much for positive comments. We, from bottom of our hearts, appreciate your questions and insightful comments. Your comments drew our attention to the many deficiencies in the details of the overall study, and also helped us to further improve the manuscript. We are glad that this research has become substantially improved under your guidance, and we hope the manuscript can be accepted for publication as soon as possible.

Point-by-point response to Reviewer #3

Question 1: Authors did experiments showed that hepatic deletion of Laptm5 increases CDC42 protein levels at PA treated conditions (Fig. 6l). What about normal condition? Authors should show results at both normal and PA treated conditions not only for Fig. 6l but also for Fig.6j-k, 6n-p. Readers and this reviewer want to see whether LPTM5 can decrease CDC42 without PA treatment. My second concern "Whether hepatic knockdown of Cdc42 ameliorates the NASH in Laptm5-HKO mice" is important. Authors should address this concern in vivo.

Response: Thank you for your advice. We agree with you that the correlation between LPTM5 and CDC42 under normal conditions is also very important and needs to be studied. *As per* your suggestions, we conducted additional

experiments to investigate the effect of LAPT5 on CDC42 at both normal and PA-treated conditions. The results showed that LAPT5 overexpression promoted the protein down-regulation of CDC42 in response to PA stimulation, but had no such effect at BSA condition (**revised fig. 6j-l, 6n-o**). Additionally, immunofluorescence co-localization staining showed that CDC42 was evenly distributed in the cytoplasm under BSA condition (**revised supplementary fig. 8f**). However, after PA treatment, CDC42 gradually moved toward lysosomes and became granular and eventually co-localized with lysosomes (**revised fig. 6p**). Collectively, these results demonstrated that the promotional effect of LAPT5 on degradation of CDC42 was PA-stimulation dependent. The results have been included in the revised manuscript (**revised fig. 6j-l, 6n-p and supplementary fig 8f**) and listed below.

figure 6

Supplementary Figure 8

As to your second concern, we agree with you that it's very important and meaningful to detect "whether hepatic knockdown of CDC42 ameliorates the NASH in *Lap^{tm5}-HKO* mice" *in vivo*. After all, it is more convincing to verify it through *in vivo* experiments. Therefore, we also hope to be able to test it *in vivo*. However, constructing the CDC42-knockdown mice in *Lap^{tm5}-HKO* background could not be accomplished within a short time. We are sorry that we could not conduct the *in vivo* experiment in this scenario. I hope you can understand the situation we are in.

In addition, to investigate the function of CDC42 in the setting of *Lap^{tm5}* deficiency in hepatocytes, we isolated primary hepatocytes from *Lap^{tm5}-KO*

mice and infected them with CDC42-knocked-down adenovirus, and subsequently detected their phenotypes. The results showed that CDC42 knockdown substantially reversed the exacerbated hepatocyte lipid deposition due to knockout of *Laptm5*. In the meantime, the activation of MAPK signaling pathway triggered by *Laptm5* deletion was consistently abolished by the knockdown of CDC42. These data clearly demonstrated that the pro-NASH capacity of *Laptm5* depletion is largely dependent on CDC42 activation. The *in vitro* results were included in the revised manuscript (**revised fig. 7e-h**) and shown below.

Question 2: Authors did not address my concern “how LAPT5 regulates MAPK signaling-related gene expression, and whether this regulation depends on CDC42”. Authors should list MAPK signaling-related genes regulated by LAPT5 and CDC42, and performed RT-qPCR assay.

Response: Thank you for your advice. To further understand the mechanism underlying the LAPT5-induced protection against NASH, we integrated the results of RNA sequencing and the Kyoto Encyclopedia of Genes and Genomes (KEGG) pathway analysis and found that *Laptm5* knockout most significantly

altered the MAPK signaling pathway, and the heatmap showed the MAPK signaling-related genes were regulated by *Laptm5* knockout (**fig. R3a**). MAPK signaling pathway include ERK, JNK and p38 pathways. However, the mRNA expression of these genes (*Mapk1*, *Mapk8* and *Mapk14*) was not altered by *Laptm5* deletion, indicating a post-translation modification of MAPK by LPTM5. Then, we examined the protein level and activity of ERK, JNK and p38, and the results exhibited that *Laptm5* deletion significantly promoted the phosphorylation levels of ERK, JNK and p38, but has no effect on their total protein levels (**revised fig. 6b-e**). It has been reported that phosphorylation of ERK, JNK, and p38 will activates downstream transcription factors such as *C-Myc*, *C-Fos*, and *JUN*, which in turn lead to expression of downstream target genes *fas*, *Tgfb1* and *Tnf* etc (*Eur Heart J.* 2019. **40(12)**:997-1008)(*Acta Pharmacol Sin.* 2022. **43(1)**. 133-145.)(*Nat Cancer.* 2021. **2(11)**. 1185-1203). To further verify whether the regulatory effect of LPTM5 on MAPK signaling depends on CDC42, we performed rescue experiments and found that CDC42 knockdown eliminated the promotional effect of *Laptm5* deletion on the activation of ERK, JNK, p38 and the expression of their downstream genes *fas*, *Tgfb1* and *Tnf* etc (**revised fig. 7e-h and fig. R3b**). Taken together, the aforementioned results showed that LPTM5 inhibited the activity of ERK, JNK, p38 and the expression of downstream molecules. The results (**revised fig. 6b-e and revised fig. 7e-h**) were included in the revised manuscript and the results (**fig. R3a and 3b**) were listed below.

figure R3a

figure 6b

figure 6c

figure 6d

figure 6e

figure 7e

figure 7f

figure 7g

figure 7h

figure R3b

Question 3: Authors did not address my concern “The molecular weight of each Western blot should be labeled clearly. SREBP-1c immunoblotting was nuclear or cytosolic form? Authors should show both forms of SREBP1c”. There are still no clear labeling in Figures 3e, 4l, s2e.

Response: Thank you for your concerns. We are so sorry that we were not able to fully understand your suggestions. And now we have clearly labeled the molecular weight of each Western blot in the revised figures, also we have arranged and uploaded uncropped versions of any gels or blots, labeled them with the relevant panels and indicated information such as the antibodies used in the supplementary files. In the meantime, both forms of SREBP-1c have also been clearly labeled in the revised manuscript (**revised fig. 3e, 4l and supplementary fig. 3e**) and are listed below.

Question 4: Authors did not address my concern “Is NEDD4L also regulated by NASH?”. Authors should measure NEDD4L protein levels in NASH livers from human patients and mouse model.

Response: Thank you for your question. According to your suggestion, we have detected NEDD4L protein levels in NASH livers from patients and mouse

models. The results showed that protein expression of NEDD4L was significantly up-regulated in NASH group compared to the control group, which proved that NEDD4L was also regulated by NASH. The results are included in the revised manuscript (**revised supplementary Fig. 2a-e**) and are shown below.

REVIEWERS' COMMENTS

Reviewer #3 (Remarks to the Author):

Authors provided new data showing that PA treatment is essential for LAPTM5 to cause CDC42 degradation. Authors should clearly stated this conclusion in the Abstract, Results, and Discussion. Authors also provided new data showing that PA treatment induced the translocation of CDC42 from cytosol to lysosome. Authors should discuss the potential molecular mechanisms.

Point-by-point response to Reviewer #3

Question 1: Authors provided new data showing that PA treatment is essential for LAPTM5 to cause CDC42 degradation. Authors should clearly stated this conclusion in the Abstract, Results, and Discussion. Authors also provided new data showing that PA treatment induced the translocation of CDC42 from cytosol to lysosome. Authors should discuss the potential molecular mechanisms.

Response: Thank you for your advice. We sincerely appreciate your work in reviewing our revised manuscript. Based on your advice, we have included the conclusions of the follow-up experiments in the manuscript and revised the content of the manuscript accordingly to present our work more clearly. The revision covers the Abstract, Results and Discussion sections of the manuscript. The revised parts included the Abstract section (lines 9-12), Results section (page 13 from the penultimate line to page 14, fourth line, and page 14 lines 16-22), Discussion section (page 18 second to last line to page 19, line 12,). The results are also listed below in bold font.

The Abstract section,

'Non-alcoholic steatohepatitis (NASH) has received great attention due to its high incidence. Here, we show that lysosomal protein transmembrane 5 (LAPTM5) is associated with NASH progression through extensive bioinformatical analysis. And protein level of LAPTM5 bears a negative

correlation with NAS score. Moreover, LAPTM5 degradation is mediated through its ubiquitination modification by the E3 ubiquitin ligase NEDD4L. Hepatocyte-specific depletion of *Laptm5* exacerbates mouse NASH symptoms. In contrast, *Laptm5* overexpression in hepatocytes exerts diametrically opposite effects. **Mechanistically, LAPTM5 interacts with CDC42 and promotes its degradation through a lysosome-dependent manner under the stimulation of palmitic acid, thus inhibiting activation of the mitogen-activated protein kinase signaling pathway.** Finally, adenovirus-mediated hepatic *Laptm5* overexpression ameliorates aforementioned symptoms in NASH models. Our results prove that LAPTM5 is an efficacious treatment for NASH and can potentially serve as a biological marker indicative of NASH progression. ’

The Results section,

‘Subsequently, the interaction between LAPTM5 and CDC42 was further corroborated by CO-IP and GST assay (**Fig. 6g, h**), and moreover, the interaction was more robust upon PA stimulation (**Fig. 6i**). **Furthermore, the *Laptm5* overexpression inhibited protein expression of CDC42 under PA stimulation, but had no such effect at BSA condition, the results being verified in both L02 cells and primary hepatocytes (Fig. 6j, k). Whereas *Laptm5* knockout promoted CDC42 expression under PA stimulation, which was also not consistent with the result at BSA condition (Fig. 6l).**

Then, we further examined the protein levels of CDC42 in the liver tissues from NASH or non-NASH individuals, and CDC42 expression was found to be significantly upregulated in the NASH group, suggesting that LAPTM5 and CDC42 were negatively correlated (**Fig. 6m**), and a NASH-regulating axis existed between LAPTM5 and CDC42.'

'Therefore, we were led to hypothesize that the down-regulation of CDC42 was mediated by the lysosomal degradation of LAPTM5. **What is more, the immunofluorescence co-localization staining exhibited that CDC42 was evenly distributed in the cytoplasm under BSA condition (Supplementary Fig. 8f). However, after PA treatment, CDC42 gradually moved toward lysosomes and became granular and eventually co-localized with lysosomes in the cells overexpressing LAPTM5, suggesting that LAPTM5 facilitated the lysosomal endocytic transport of CDC42 and promoted its degradation by lysosomes (Fig. 6p).'**

The Discussion section,

'Prior studies have proved that TMBIM1 promoted lysosomal degradation of TLR4 and inhibited high-fat diet-induced insulin resistance, hepatic steatosis and inflammation(40). **In the present study, we found that LAPTM5 could promote the lysosomal localization and degradation of CDC42 under the stimulation of PA. However, the expression and localization of CDC42**

were not influenced under the BSA condition. Which indicates that the correlation between LAPTM5 and CDC42 may have changed after the treatment of PA. For example, there may be some changes in the protein domains and molecular activity of LAPTM5 and CDC42 after PA stimulation. Youngshil et al. reported that the protein trafficking and sorting functions of LAPTM5 were strictly regulated by different domains of it(27). Manju et al. reported that the activation of CDC42 plays a key role in the regulation of disease progression, and the activity intensity of CDC42 also affects the biological function of it(26). Therefore, after PA stimulation, delicate and complex changes may have occurred inside the cells. And these deep-seated mechanisms require further study.'

We are very grateful for your advice which serves as a guidance for the the further improvement of the manuscript, and hope that our study can be successfully accepted for publication with the revisions.